

# Parity effects and universal terms of $\mathcal{O}(1)$ in the entanglement near a boundary

Henning Schlömer[1,2⋆], Chunyu Tan[3], Stephan Haas[3] and Hubert Saleur[3,4]

**1** Department of Physics and Arnold Sommerfeld Center for Theoretical Physics (ASC),
Ludwig-Maximilians-Universität München, München D-80333, Germany
**2** Munich Center for Quantum Science and Technology (MCQST),
D-80799 München, Germany
**3** Department of Physics and Astronomy, University of Southern California,
Los Angeles, California 90089-0484, USA
**4** Institut de physique théorique, CEA, CNRS, Université Paris-Saclay

⋆ H.Schloemer@physik.uni-muenchen.de

## Abstract

In the presence of boundaries, the entanglement entropy in lattice models is known to exhibit oscillations with the (parity of the) length of the subsystem, which however decay to zero with increasing distance from the edge. We point out in this article that, when the subsystem starts at the boundary and ends at an impurity, oscillations of the entanglement (as well as of charge fluctuations) appear which do not decay with distance, and which exhibit universal features. We study these oscillations in detail for the case of the XX chain with one modified link (a conformal defect) or two successive modified links (a relevant defect), both numerically and analytically. We then generalize our analysis to the case of extended (conformal) impurities, which we interpret as SSH models coupled to metallic leads. In this context, the parity effects can be interpreted in terms of the existence of non-trivial topological phases.



## 1   Introduction

Quantum entanglement is at the heart of quantum mechanics, its diverse theoretical interest covering topics from quantum information science [1] to condensed matter [2,3] and high energy physics [4,5]. The entanglement entropy $S_A$ of spatial region $A$ is given by the von Neumann entropy of the reduced density matrix $\rho_A$, computed by integrating out the degrees of freedom located in the complement of $A$,

$$S_A = -\mathrm{Tr}_A(\rho_A \ln \rho_A). \tag{1}$$

For $1+1$ dimensional conformal field theories (CFTs), it is possible to derive universal predictions for the entanglement entropy (see e.g. [6] for a review)

$$S_A = \frac{c}{3} \ln\left(\frac{\ell}{a}\right) + c_1', \tag{2}$$

where $c$ is the central charge of the CFT, $A$ is a single interval of length $\ell$, $a$ is a UV cutoff, and we dropped terms which vanish as $a \to 0$. While the logarithmically divergent term is governed only by the central charge $c$ of the CFT, the constant term, $c_1'$, is non-universal, and can be changed by a redefinition of the cutoff.

    In the presence of a boundary, however, the constant term becomes more relevant, as the dependence on the UV cutoff can be eliminated. If $A$ denotes again a single interval of length $\ell$, now starting at the boundary of a semi-infinite system, one finds [6]

$$S_{A,\mathrm{bdr}} = \frac{c}{6} \ln\left(\frac{2\ell}{a}\right) + \tilde{c}_1', \tag{3}$$

with the same cut-off $a$ as in Eq. (2), along with

$$\tilde{c}_1' - \frac{c_1'}{2} = \ln g, \tag{4}$$

where $\ln g$ is the Affleck-Ludwig boundary entropy [7]. Note the factor of 2 in the argument of the logarithm in Eq. (3), first pointed out in [8].

    Terms of $\mathcal{O}(1)$ are also physically meaningful for CFTs with topological defects [9–13]. The case where such defects are inside the interval is well understood, and closely related to the boundary case (via folding). The case where the topological defect is at the border of the interval[1] is more controversial [14,15]. Here, we report on yet other situations where universal terms of $\mathcal{O}(1)$ do in fact appear, with interesting physical interpretations.

---

[1]To avoid confusion, we use the words boundary and edge to refer to physical boundaries, and the word border to refer to the extremity or extremities of the sub-systems.

The first such situation occurs when combining a conformal defect with a boundary[2]. Specifically, we consider an XX chain with a modified bond (sometimes also referred to as impurity bond) of value $J'$ (the value of the bulk bonds is $J$). In the bulk, the entanglement of a region of length $\ell$ ending on the modified bond is known to take the form [17]

$$S_{A,\text{imp}} \approx \frac{(c_{\text{eff}}(\lambda)+c)}{6}\ln\frac{\ell}{a} + \frac{(c_1'(\lambda)+c_1')}{2}\,, \tag{5}$$

where we have set

$$\lambda \equiv \frac{J'}{J}\,. \tag{6}$$

In Eq. (5), $c_{\text{eff}}(\lambda)$ is an "effective central charge" whose expression is known analytically, interpolating between $c_{\text{eff}}(0)=0$ and $c_{\text{eff}}(1)=c$. The term of $\mathcal{O}(1)$ also depends on $\lambda$, with $c_1'(\lambda=1)=c_1'$. There are no parity effects at large distances.

In this article, we combine a conformal defect with a boundary by considering the XX chain with free boundary conditions, and choosing the interval of length $\ell$ so that it starts at the boundary and ends on the modified bond. As discussed below, we find that the entanglement entropy now exhibits strong parity effects [3], and that one can write[4]

$$\begin{aligned} S_{A,\text{imp+bdr}}^e &= \frac{c_{\text{eff}}(\lambda)}{6}\ln\left(\frac{2\ell}{a}\right) + \tilde{c}_1'^e(\lambda)\,, \\ S_{A,\text{imp+bdr}}^o &= \frac{c_{\text{eff}}(\lambda)}{6}\ln\left(\frac{2\ell}{a}\right) + \tilde{c}_1'^o(\lambda)\,, \end{aligned} \tag{7}$$

where superscripts $e$ and $o$ denote the cases where the size of subsystem $A$ is even and odd, respectively. The surprising observation is that the parity effects in Eq. (7) do not decay with $\ell$ but remain of order one. In particular,

> We conjecture that $\tilde{c}_1'^e(\lambda)-\tilde{c}_1'^o(\lambda) \equiv \delta S$ is a $\ell$ independent universal function of the phase shift at the Fermi surface.

Here, by universal we mean that this term does not depend on the cutoff (in contrast to e.g. $c_1'$), and that it can be calculated using field theory, where it depends on a single parameter - that can be traded with the phase shift at the Fermi surface.

While parity effects for the entanglement entropy are well known to occur in the presence of a boundary[5] [19, 20], these effects, without an impurity, decay with large $\ell$ (like $\ell^{-1}$ for the XX chain[6] [21]).

We further study the case when the impurity induces an RG flow. Specifically, we consider the case of an XX model with two successive modified bonds, giving rise to a resonant level or "dot". The symmetry of this problem is different from the case of one impurity, and as a result the defect is not conformal any longer. Most interesting is the limit where $J' \to 0$, where universal "healing" is observed, with the chain appearing homogeneous at distances

---

[2]The case of topological defects in the presence of boundaries can be handled by conformal methods [16].

[3]A different but somewhat related observation was made earlier in [18].

[4]Here - as in all similar statements below - our results apply to the limits $\ell, L \to \infty$ (with $\frac{\ell}{L}$ fixed), that is, in the scaling limit. Most results would not hold in finite size.

[5]Parity effects are known to occur in the bulk as well for general Rényi entropies.

[6]The entropy for the XXZ chain exhibits strong, oscillating corrections to the conformal result, decaying with an amplitude $\ell^{-K}$, where $K$ is the Luttinger liquid exponent (the dimension of the most relevant operator in the bulk theory). The XX case has been studied in considerable detail [21], with the result that

$$S_{A,\text{bdr}} = \frac{1}{12}\ln\ell + \mathcal{O}(1) + f_1\sin[(2\ell+1)k_F]\ell^{-1} + o(\ell^{-1})\,.$$

Here, $f_1$ is a non-universal constant, $k_F$ is the Fermi momentum, so for $k_F = \frac{\pi}{2}$ (no magnetic field) we get the usual oscillating term $(-1)^\ell$.

large compared to a crossover length scale $\frac{1}{T_B} \propto \frac{1}{J'^2}$. For an interval in the bulk that ends on the dot on one side (or actually on a dot on either side) the entanglement obeys a scaling relation of the type [22]

$$\frac{dS_{A,\text{imp}}}{d\ln\ell} = G(\ell T_B), \tag{8}$$

where $G$ is a sort of running effective central charge. Note that in Eq. (8) we considered only the derivative of the entanglement with respect to $\ell$, not $S$ itself, since the latter quantity involves several non-universal terms of $\mathcal{O}(1)$ because of the additional $T_B$ dependency [22]. Once again, we can ask what happens in the presence of such an impurity combined with a boundary as before. We will then see that

$$\frac{dS^e_{A,\text{imp+bdr}}}{d\ln\ell} = F(\ell T_B) + f^e(\ell T_B),$$
$$\frac{dS^o_{A,\text{imp+bdr}}}{d\ln\ell} = F(\ell T_B) + f^o(\ell T_B), \tag{9}$$

where $F, f^o, f^e$ are non-trivial, universal functions[7]. The difference,

$$f^e(\ell T_B) - f^o(\ell T_B) \equiv \delta\frac{dS(\ell T_B)}{d\ln\ell}, \tag{10}$$

turns out to exhibit a most interesting crossover between the UV ($\ell T_B \ll 1$) and the IR ($\ell T_B \gg 1$) regimes.

It has often been observed in the past that charge fluctuations exhibit features qualitatively similar to those of the entanglement [23, 24], while being easier to handle technically and more realistic to measure in an experiment. As we will see below, this is still the case in our problem, where the qualitative physics of universal parity effects of $\mathcal{O}(1)$ is the same as for the entanglement entropy.

This article is organized as follows. In Section 2, we analyze the XX chain with a single modified bond, and establish strong evidence for the aforementioned conjectures. In particular, we study the von-Neumann entropy in 2.1 and the charge fluctuations in 2.2. In Section 3, we present analytical insights into the problem in the continuum limit and obtain perturbative results in the regime $\lambda \lesssim 1$. We briefly study extended conformal impurities with several alternating modified bonds in Sec. 4, yielding interesting physical insights into the single impurity model in terms of topological phases of the SSH model. Finally, we consider the resonating quantum dot model in Sec. 5. Conclusions and possible extensions of our results are discussed in Sec. 6.

## 2 Single impurity link: numerical study

We start by analyzing non-decaying parity effects for both the entanglement entropy and charge fluctuations in models with a single conformal defect located at the border of the subsystem, as introduced in Sec. 1.

### 2.1 Entanglement entropy

To our knowledge, the function $c_{\text{eff}}(\lambda)$ in Eq. (5) has been determined analytically only in the bulk case of an XX chain with impurity bond, with the result [11, 12, 17]

$$c_{\text{eff}} = -\frac{6}{\pi^2}\Big\{[(1+s)\ln(1+s) + (1-s)\ln(1-s)]\ln s + (1+s)\text{Li}_2(-s) + (1-s)\text{Li}_2(s)\Big\}. \tag{11}$$

---

[7]It is not so clear how $F$ is related to $G$ in the bulk because of potential interactions between the two borders of the interval.

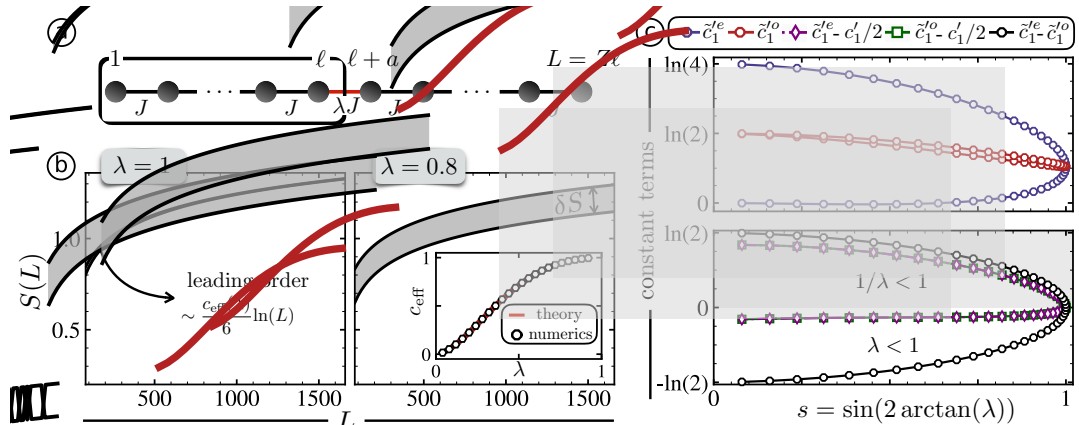

Figure 1: Entanglement entropy in a tight binding system with a single impurity link and open boundaries. (a) Illustration of the model. The bond placed between sites at $\ell$ and $\ell + a$ (corresponding to site indices $j_0$ and $j_0 + 1$, respectively) is modified, and has amplitude $\lambda J$. The total system size is set to be $L = Z\ell$. (b) Entanglement entropy of subsystem $A$ (boxed region in (a)) with $B$ (non-circled region in (a)) for $\lambda = 1$ (left) and $\lambda = 0.8$ (right) as a function of the total system size $L = Z\ell$, with $Z$ fixed to $Z = 10$. For $\lambda < 1$, non-decaying parity effects of $\mathcal{O}(1)$ are observed, illustrated by the shaded region. The inset shows the predicted central charges $c_{\text{eff}}(\lambda)$ compared to the numerically extracted leading order behavior for varying impurity strength. (c) Even component $\tilde{c}_1'^{e}(s)$ (upper panel, blue circles) and odd component $\tilde{c}_1'^{o}(s)$ (upper panel, red circles) of the entanglement, the corresponding difference $\tilde{c}_1'^{e}(s) - \tilde{c}_1'^{o}(s)$ (lower panel, black circles) and the isolated boundary parts $\tilde{c}_1'^{e/o}(s) - c_1'(s)/2$ (lower panel, purple diamonds and green squares). $\tilde{c}_1'^{e}(s)$, $\tilde{c}_1'^{o}(s)$ and $c_1'(s)/2$ are obtained by numerically fitting the data to Eq. (21) for open and Eq. (20) for periodic boundaries.

Here,

$$\text{Li}_2(z) = -\int_0^z dx \frac{\ln(1-x)}{x}, \tag{12}$$

and we have introduced the variable

$$s = \sin(2\arctan\lambda) = \frac{2JJ'}{J^2 + (J')^2}. \tag{13}$$

We recall that $J'$ is the modified link, and $\lambda = \frac{J'}{J}$. With the identity

$$\text{Li}_2(s) = \frac{\pi^2}{6} - \text{Li}_2(1-s) - \ln s \ln(1-s), \tag{14}$$

we can rewrite Eq. (11) as

$$c_{\text{eff}} = s - 1 - \frac{6}{\pi^2}\left\{[(1+s)\ln(1+s)\ln s + (1+s)\text{Li}_2(-s) + (s-1)\text{Li}_2(1-s)\right\}, \tag{15}$$

an expression which is also often found in the literature. We note that $c_{\text{eff}}$ is rather well approximated by $c_{\text{eff}} \approx s^2$. Furthermore, from Eq. (13), it is evident that the variable $s$ obeys the symmetry

$$s(\lambda) = s\left(\frac{1}{\lambda}\right), \tag{16}$$

and therefore, $c_{\text{eff}}$ being analytical in $s$, we have

$$c_{\text{eff}}(\lambda) = c_{\text{eff}}\left(\frac{1}{\lambda}\right). \tag{17}$$

We also note (see Sec. 3.2 for details) that the variable $s$ has a physical meaning: $s = \cos\xi$, where $\xi$ is the phase shift at the Fermi surface[8],

$$|\xi(\lambda)| \equiv \frac{\pi}{2} - 2\arctan\lambda, \ \lambda \in [0, 1]. \tag{18}$$

It is known that $c_{\text{eff}}$ depends only on this phase shift, and not on the microscopic mechanism (such as the exact realization of the "impurity") giving rise to it, see e.g. [25].

In the following, we consider a non-interacting, one-dimensional chain of hopping electrons, cf. Fig. 1 ⓐ for an illustration. Specifically, we consider a system of size $L$ (with $N = L/a$ sites and $a$ the lattice spacing), which is fixed by the size of subsystem $A$, i.e., $\ell$, via $L = Z\ell$, where $Z \in \mathbb{N}$. We introduce a single impurity link of strength $J' = \lambda J$ at the border of subsystem $A$ at site $j_0 = \ell/a$. The corresponding Hamiltonian is given by

$$H = -J \sum_{j=1}^{N} \left(c_{j+1}^{\dagger}c_j + c_j^{\dagger}c_{j+1}\right) - J(\lambda - 1)\left(c_{j_0+1}^{\dagger}c_{j_0} + c_{j_0}^{\dagger}c_{j_0+1}\right). \tag{19}$$

By first using periodic boundary conditions, we check that the entanglement is given by the natural generalization of Eq. (5) for finite sizes, i.e.,

$$S_{A,\text{imp}} \approx \frac{(c_{\text{eff}}(\lambda) + c)}{6}\ln\left(\frac{L}{\pi a}\sin\frac{\pi\ell}{L}\right) + \frac{(c_1'(\lambda) + c_1')}{2}, \tag{20}$$

where we evaluate the entanglement entropy via the correlation matrix of the tight-binding model [26]. Moreover, we verify that the parity effects decay with increasing size, as expected in a system with closed boundaries. This is true more precisely when $\ell, L$ become large, with fixed ratio $Z = L/\ell$ (including the case $\ell \ll L$). In the following, we fix $Z = 10$, i.e., the size of subsystem $A$ is 10% of the total system size, with the modified bond located at the right border. An example of the decay of oscillatory behavior is illustrated in Fig. 2. The difference of terms of $\mathcal{O}(1)$ between the even and odd cases, denoted by $\delta S$, is found to vanish as $\ell^{-1}$ (see inset of Fig. 2), in agreement with standard predictions for the XX chain.

Near a boundary, the slope of the entanglement is divided in half, just like for ordinary, homogeneous systems. We note that there doesn't seem to be a simple derivation of this result in the literature, in particular when finite size effects are taken into account. We have checked numerically that Eq. (7) gets modified as announced in the introduction, i.e.,

$$S_{A,\text{imp+bdr}}^{e} = \frac{c_{\text{eff}}(\lambda)}{6}\ln\left(\frac{2L}{\pi a}\sin\frac{\pi\ell}{L}\right) + \frac{A^e(\lambda)}{\ell} + \tilde{c}_1'^e(\lambda),$$
$$S_{A,\text{imp+bdr}}^{o} = \frac{c_{\text{eff}}(\lambda)}{6}\ln\left(\frac{2L}{\pi a}\sin\frac{\pi\ell}{L}\right) + \frac{A^o(\lambda)}{\ell} + \tilde{c}_1'^o(\lambda), \tag{21}$$

where we explicitly take into account the leading order ($\lambda$ dependent) decaying terms $\sim \ell^{-1}$ for a better quality of the numerical fits. Numerical results for $c_{\text{eff}}$ are presented in the inset of Fig. 1 ⓑ, in excellent agreement with the predicted value, Eq. (11). The crucial point is now that the parity effects *do not decay with* $L$. This is illustrated in detail in Fig. 1 ⓑ, where the entanglement entropy is shown when increasing $L$ for fixed $Z = 10$. In particular, the left panel shows the case $\lambda = 1$, in which case oscillations decay as $\sim L^{-1}$. However, when $\lambda \neq 1$,

---

[8]The sign of $\xi$ is discussed below.

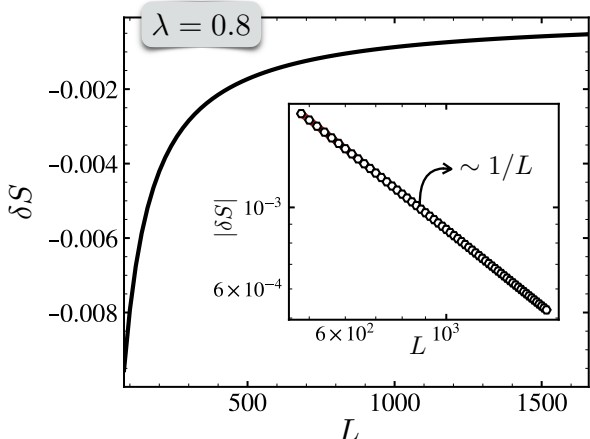

Figure 2: Entanglement entropy difference $\delta S$ between subsystem size even and odd when $\lambda = 0.8$. To simulate bulk effects, we apply periodic boundary conditions, and in particular choose $Z = 10$. The entanglement differences are seen to scale as $\sim 1/L$, as depicted in the double logarithmic plot in the inset.

finite oscillations persist and saturate towards the value $\delta S(\lambda)$ for $L \to \infty$, illustrated by the grey shaded region in Fig. 1 (b).

We numerically evaluate the entanglement entropies for varying impurity strengths $\lambda$ (where we allow both $\lambda < 1$ and $\lambda > 1$), and extract the values of $\tilde{c}_1^{\prime e/o}(\lambda)$ by fitting the data to Eq. (21). The upper panel of Fig. 1 (c) shows the even and odd contributions $\tilde{c}_1^{\prime e/o}(\lambda)$ as a function of $s = \sin(2 \arctan(\lambda))$, featuring strong and weak dependencies on the tunable impurity strength in the even and odd case, respectively. Remarkably, the difference between these two cases, $\delta S(\lambda) = \tilde{c}_1^{\prime e}(\lambda) - \tilde{c}_1^{\prime o}(\lambda)$ (see the black curve in the lower panel of Fig. 1 (c)), is a seemingly simple function of $s$, exhibiting the symmetry

$$\delta S(s) = -\delta S(-s), \tag{22}$$

or, equivalently,

$$\delta S(\lambda) = -\delta S\left(\frac{1}{\lambda}\right). \tag{23}$$

Like the effective central charge $c_{\text{eff}}$, we expect $\delta S$ to be a function of the phase shift at the Fermi surface only. Our curves $\delta S(s)$ should thus be the same for every similar problem once $s = \cos \xi$ is being used. Examples of this will be given below, in particular when we extend to impurities that include several modified bonds in Sec. 4.

We further note that $\delta S(s)$ seems to be well approximated by a simple ellipse with semi-major (-minor) axis 1 (ln 2),

$$\delta S \approx \pm \ln 2 \sqrt{1 - s^2}, \tag{24}$$

which, however, turns out to be not exact - a fact that can be proven analytically, as discussed in Sec. 3.1.

In analogy to Eq. (4), we further isolate the boundary contribution of the entanglement entropy,

$$\tilde{c}_{\text{bdry}}^{\prime e/o}(\lambda) = \tilde{c}^{\prime e/o}(\lambda) - \frac{c_1'(\lambda)}{2}, \tag{25}$$

where we compute $c_1'(\lambda)$ by fitting the numerical outcome of an impurity system with periodic boundary conditions to Eq. (20). The boundary contributions $\tilde{c}_{\text{bdry}}^{\prime e/o}(\lambda)$ are observed to lie on

Table 1: Simplified valence bond picture for strong (weak) impurity bonds $\lambda \ll 1$ ($1/\lambda \ll 1$) for both even and odd scenarios. The bonds separating the regions of the bipartition are illustrated by grey lines, and inter- (intra-) subsystems valence bonds are depicted by black (dark blue) curved lines. Each blue line corresponds to an entanglement entropy of $\ln 2$.

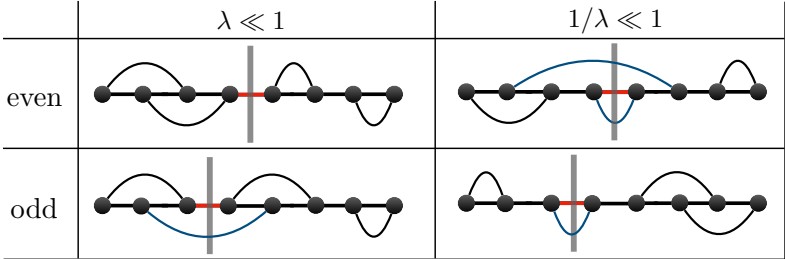

the same curve, see the lower panel of Fig. 1 ⓒ - with the subtle distinction that

$$\tilde{c}_{\mathrm{bdry}}^{\prime p}(\lambda) = \tilde{c}_{\mathrm{bdry}}^{\prime \bar{p}}(1/\lambda), \tag{26}$$

where $\bar{o} = e, \bar{e} = o$. Indeed, from Eq. (26), the (anti-)symmetry of $\delta S(\lambda)$ in Eq. (23) immediately follows.

The limiting cases of strong and weak modified bonds can be intuitively interpreted in a simplified valence bond picture [20], the different scenarios in the even and odd cases depicted in Tab. 1. For $\lambda \ll 1$, it is unfavorable to form a valence bond over the impurity. This, in turn, leads to an additional contribution to the entanglement entropy between the two subsystems on the left and right of the impurity of $\ln 2$ in the odd case compared to the even case. For a strong impurity bond, $1/\lambda \ll 1$, on the other hand, it is highly favorable to form a valence bond over the impurity. Focusing first on the even case, this leaves an odd number of sites on both sides, that have to be "assigned" a valence bond partner, leading to an extra valence bond between the two subsystems - and hence an additional $\ln 2$ entanglement entropy compared to the odd case. Note that the valence bond picture does not represent the exact ground state of the impurity chain, however it provides an intuitive understanding of the observed entanglement entropy differences in the limits of weak and strong impurities, cf. Fig. 1 ⓒ.

## 2.2 Charge fluctuations

As mentioned in the introduction, fluctuations usually behave in a way that is qualitatively similar to entanglement [23]. Although there is, to our knowledge, no precise, universal version of this statement, many examples have been found, especially in the non-interacting case. As we will show, our system is no exception. We thus consider the same model as before, but now instead of the entanglement of the subsystem situated between the boundary and the impurity, we consider the fluctuations of the number of electrons it contains.

First, with no impurity and in the bulk, it is well known that the charge fluctuations in the non-interacting case - up to decaying, non-universal parity terms - take the form

$$\langle (Q - \langle Q \rangle)^2 \rangle_A = \mathcal{F}_A(\ell) = \frac{C}{\pi^2} \ln \frac{\ell}{a} + C_1', \tag{27}$$

where the coefficient $C$ takes the value $C = 1$, while $C_1'$ is non-universal, and changes as usual under cutoff re-definitions[9]. We emphasize that $C$ is not related with the central charge: with

---

[9]For the XX chain, and the cutoff given by the lattice spacing, one can show that $C_1' = \frac{1+\gamma+\ln 2}{\pi^2}$, where $\gamma$ is Euler's constant [23].

interactions of Luttinger type for instance, $C$ would vary while $c = 1$ is a constant. In the presence of an impurity and in the bulk, one can also show (see Sec. 3.2) that

$$\mathcal{F}_{A,\text{imp}} = \frac{C_{\text{eff}}(\lambda) + 1}{2\pi^2} \ln \frac{\ell}{a} + \frac{(C_1'(\lambda) + C_1')}{2}, \tag{28}$$

with

$$C_{\text{eff}} = \cos^2 \xi = s^2. \tag{29}$$

Finally, near a boundary but without impurity, one has [23]

$$\mathcal{F}_{A,\text{bdr}} = \frac{1}{2\pi^2} \ln \frac{2\ell}{a} + \frac{C_1'}{2}. \tag{30}$$

Note the similarity of all these results with those for the entanglement entropy. This carries over to finite size effects as well - for instance, equation (30) when the system has total length $L$ becomes

$$\mathcal{F}_{A,\text{bdr}} = \frac{1}{2\pi^2} \ln \left( \frac{2L}{\pi a} \sin \frac{\pi \ell}{L} \right) + \frac{C_1'}{2}. \tag{31}$$

Parity effects are also known to occur for the fluctuations. Like in the case of the entanglement, they decay with $\ell$ large[10]. We shall see now that, in the presence of both the impurity and the boundary, we have[11]

$$\mathcal{F}_{A,\text{imp+bdr}}^e = \frac{C_{\text{eff}}(\lambda)}{2\pi^2} \ln \left( \frac{2L}{\pi a} \sin \frac{\pi \ell}{L} \right) + \frac{A^e(\lambda)}{\ell} + B^e(\lambda) \frac{\ln \ell}{\ell} + \tilde{C}_1'^e(\lambda),$$
$$\mathcal{F}_{A,\text{imp+bdr}}^o = \frac{C_{\text{eff}}(\lambda)}{2\pi^2} \ln \left( \frac{2L}{\pi a} \sin \frac{\pi \ell}{L} \right) + \frac{A^o(\lambda)}{\ell} + B^o(\lambda) \frac{\ln \ell}{\ell} + \tilde{C}_1'^o(\lambda). \tag{32}$$

For the single impurity link model, we numerically evaluate the correlator $G_{ij} = \langle c_i^\dagger c_j \rangle$, from which the density-density correlations can be computed,

$$\langle c_i^\dagger c_i \, c_j^\dagger c_j \rangle = \langle c_i^\dagger c_i \rangle \langle c_j^\dagger c_j \rangle + \langle c_i^\dagger c_j \rangle \langle c_i \, c_j^\dagger \rangle = \langle c_i^\dagger c_i \rangle \langle c_j^\dagger c_j \rangle + \langle c_i^\dagger c_j \rangle \left( \delta_{ij} - \langle c_j^\dagger c_i \rangle \right). \tag{33}$$

The charge fluctuations in a given interval of length $\ell$ are then evaluated via

$$\mathcal{F}(\ell) = \sum_{i,j=1}^{\ell} \langle n_i n_j \rangle - \langle n_i \rangle \langle n_j \rangle = \sum_{i,j=1}^{\ell} \langle c_i^\dagger c_j \rangle \left( \delta_{ij} - \langle c_i^\dagger c_i \rangle \right). \tag{34}$$

Note that the open boundary, non-interacting model with an impurity can also be solved semi-analytically using a plane-wave ansatz that scatters at the impurity. The plane wave amplitudes as well as the condition for the momentum quantization can then be solved numerically using the scattering ansatz together with the boundary conditions, from which the fluctuations can be computed. In our numerical results however, we stick to the same strategy as used for

---

[10]For the XX model for instance, the leading decaying terms for a homogeneous system with a boundary read

$$\mathcal{F}_{A,\text{bdr}} = \frac{1}{2\pi^2} \ln \frac{2\tilde{\ell}}{a} + \frac{C_1'}{2} + \frac{1}{2\pi^2(2\tilde{\ell})} - \frac{(-1)^\ell}{\pi^2(2\tilde{\ell})} \left[ \ln 2\tilde{\ell} + \gamma + \ln 2 \right] + \mathcal{O}(\ell^{-2}),$$

where $\tilde{\ell} = (L/a\pi) \sin \pi\ell/L$. Note the similarity with the result for the entanglement entropy, however with the additional oscillating contribution $\propto (-1)^\ell \ln \tilde{\ell}/\tilde{\ell}$. This additional oscillating term has previously been analyzed for Luttinger liquids (LL), where amplification (suppression) of the decaying parity effects have been observed for LL parameters $K < 1$ ($K \geq 1$) [23].

[11]The coefficients $A^o, A^e$ are not the same as those appearing in the entanglement. We kept the same notation for simplicity.

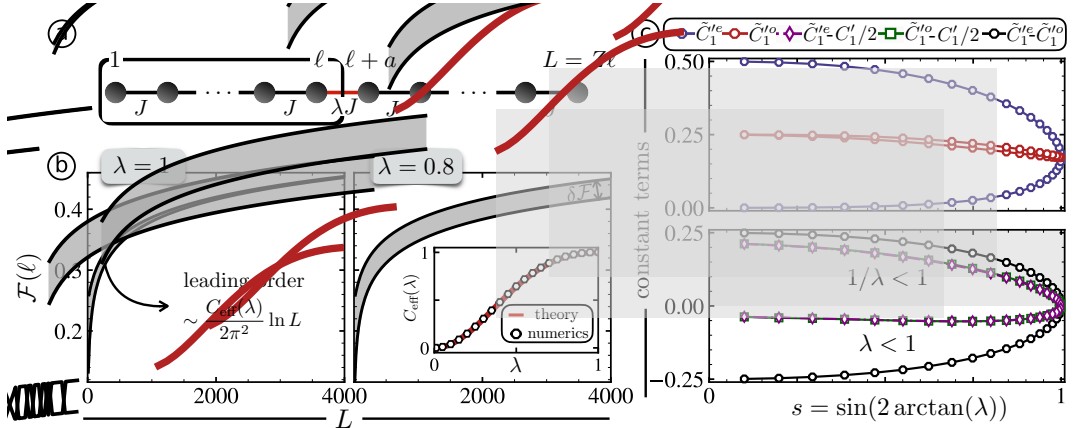

Figure 3: Charge fluctuations in a tight binding system with a single impurity link and open boundaries. (a) Illustration of the model. An impurity is placed between sites at $\ell$ and $\ell + a$ (corresponding to site indices $j_0$ and $j_0 + 1$, respectively), with impurity hopping $\lambda J$. The total system size is set to be $L = Z\ell$, where we choose $Z = 10$. (b) Fluctuations in subsystem $A$ (circled region in (a)) for $\lambda = 1$ (left) and $\lambda = 0.8$ (right). For $\lambda < 1$, non-decaying parity effects of $\mathcal{O}(1)$ are observed, illustrated by the gray background. The inset shows the predicted values of $C(\lambda)$, cf. Eq. (27), compared to the numerically extracted leading order behavior for varying impurity strength. (c) Even component $\tilde{C}_1'^e$ (upper panel, blue circles) and odd component $\tilde{C}_1'^o$ (upper panel, red circles) of the charge fluctuations, the corresponding difference $\tilde{C}_1'^e - \tilde{C}_1'^o$ (lower panel, black circles) and the isolated boundary parts $\tilde{C}'^{e/o} - C_1'/2$ (lower panel, purple diamonds and green squares) for varying $\lambda$. $\tilde{C}_1'^e(\lambda)$, $\tilde{C}_1'^o(\lambda)$ and $C_1'(\lambda)/2$ are obtained by numerically fitting the data to Eq. (32) for open and Eq. (35) for periodic boundary conditions.

evaluating the entanglement entropy, that is we use numerical diagonalization of the tight-binding matrix, compute the fluctuations and fit the data to Eq. (32). In Sec. 4, we use the plane-wave ansatz to exemplify that the physics is governed solely by the properties of the system at the Fermi level.

Our numerics are presented in Fig. 3. Akin to the entanglement entropy, we observe strong parity effects of $\mathcal{O}(1)$ for non-trivial impurity bonds $\lambda \neq 1$, see i.p. Fig. 3 (b). The parameter $C_{\text{eff}}(\lambda)$, which plays the role of the effective central charge and which we calculate from a generalization of Eq. (32) similar to what has been done for the entanglement entropy, is further verified to coincide with analytical predictions (see Sec 3.2 for details). Furthermore, the fitted even (odd) components $\tilde{C}'^e(s)$ ($\tilde{C}'^o(s)$) of the $\mathcal{O}(1)$ contributions to the fluctuations vary strongly (slightly) with the impurity strength $\lambda$, but collapse onto a symmetric curve when considering their difference $\delta\mathcal{F}(\lambda) = \tilde{C}'^e(\lambda) - \tilde{C}'^o(\lambda)$.

Note that for the charge fluctuations, the relevant scale is given by $1/4$ (in comparison to $\ln 2$ for the entanglement), which can be understood by considering a mapping from the non-interacting tight binding chain to the XX spin model, where $\langle S_i^z S_i^z \rangle = 1/4$ at half-filling, i.e., without a magnetic field.

We further extract the constant term for a system with periodic boundaries, where we again explicitly take into account the leading order parity terms, i.e., we fit the impurity system with periodic boundaries to

$$\mathcal{F}_{A,\text{imp}} = \frac{C_{\text{eff}}(\lambda) + 1}{2\pi^2} \ln\left(\frac{2L}{\pi a} \sin\frac{\pi\ell}{L}\right) + \frac{A(\lambda)}{\ell} + B(\lambda)\frac{\ln\ell}{\ell} + \frac{(C_1'(\lambda) + C_1')}{2}. \tag{35}$$

Computing $C'_1(\lambda)$, we can calculate the isolated boundary contributions to the fluctuations, $\tilde{C}'^{e/o}_{\text{bdry}}(\lambda) = \tilde{C}'^{e/o}(\lambda) - C'_1(\lambda)/2$. Numerically, we again observe that

$$\tilde{C}'^p_{\text{bdry}}(\lambda) = \tilde{C}'^{\bar{p}}_{\text{bdry}}(1/\lambda), \tag{36}$$

see the lower panel of Fig. 3 ©. From Eq. (36) immediately follows the symmetry of $\delta\mathcal{F}(\lambda)$,

$$\delta\mathcal{F}(\lambda) = -\delta\mathcal{F}(1/\lambda). \tag{37}$$

# 3 Single impurity link: analytical insights

After this detailed, phenomenological study, we now turn to some analytical considerations. In particular, we present a general analysis of the scaling limit together with perturbative calculations, both for the entanglement entropy in Sec. 3.1 and fluctuations in Sec. 3.2[12].

## 3.1 Entanglement entropy

We start by recalling the formulas for the decomposition of lattice fermions into continuous fields,

$$c_j \mapsto e^{ik_F j}\psi_R + e^{-ik_F j}\psi_L. \tag{38}$$

At half-filling, $k_F = \frac{\pi}{2}$. For a chain starting at $j = 1$, to take into account the chain termination we have formally $c_{j=0} = 0$, so at that extremity the boundary conditions read $\psi_L = -\psi_R$. We use the bosonization formulas (see e.g. [27])

$$\psi_R = \frac{1}{\sqrt{2\pi}}\eta_R e^{i\sqrt{4\pi}\phi_R}; \ \psi_L = \frac{1}{\sqrt{2\pi}}\eta_L e^{-i\sqrt{4\pi}\phi_L}, \tag{39}$$

where $\eta_{R,L}$ are anti-commuting cocycles (Klein factors) necessary to ensure anti-commutation of the fermion fields[13]. We see that the boundary conditions correspond to Dirichlet boundary conditions, i.e.,

$$\phi(x = 0, t) \equiv \phi_R + \phi_L = \left(n + \frac{1}{2}\right)\sqrt{\pi}. \tag{40}$$

If we consider a homogeneous chain starting at $j = 1$ with a single modified link at $j_0$ with Hamiltonian[14]

$$H = -J\sum_{j=1}^{\infty}\left(c^{\dagger}_{j+1}c_j + c^{\dagger}_j c_{j+1}\right) - J(\lambda - 1)\left(c^{\dagger}_{j_0+1}c_{j_0} + c^{\dagger}_{j_0}c_{j_0+1}\right), \tag{41}$$

the bosonized form of the perturbation reads[15]

$$c^{\dagger}_{j_0+1}c_{j_0} + c^{\dagger}_{j_0}c_{j_0+1} \mapsto \frac{(-1)^{j_0}}{2\pi}\left(i\eta_R\eta_L e^{-i\sqrt{4\pi}\phi(j_0 a)} - i\eta_L\eta_R e^{i\sqrt{4\pi}\phi(j_0 a)} + \text{h.c.}\right) =$$
$$\frac{(-1)^{j_0}}{\pi}2i\eta_R\eta_L\cos\sqrt{4\pi}\phi(j_0 a). \tag{42}$$

---

[12]Unfortunately, we have not been able to derive the exact form of the curves $\delta S(\lambda)$ or $\delta\mathcal{F}(\lambda)$.

[13]We use the convention that $\phi_R, \phi_L$ commute; we also use non normal-ordered exponentials, so there is no need for the usual $1/\sqrt{a}$ factor.

[14]We assume that the Hamiltonian is properly normalized so that its continuum limit is relativistically invariant, $J = \frac{1}{2}$.

[15]We have only retained the leading contribution, which is exactly marginal in the RG sense. All higher order terms are irrelevant, and should not contribute in the limit $\ell \gg 1$.

We have used that $\eta_{R,L}^{\dagger} = \eta_{R,L}$ and $\eta_{R,L}^2 = 1$. Note that $(i\eta_R\eta_L)^2 = -\eta_R\eta_L\eta_R\eta_L = 1$; in what follows we will treat the term $i\eta_R\eta_L$ as a number (say 1)[16]. The crucial point is that the perturbation in the field theory has a component whose sign differs between the cases $j_0$ even and $j_0$ odd.

We can now attempt to carry out a perturbative analysis of the entanglement in the limit of weak impurities $1 - \lambda \ll 1$. In the continuum limit, the Hamiltonian reads (we take $J = \frac{1}{2}$ to ensure a relativistic dispersion relation of low energy excitations with velocity unity)

$$H = \frac{1}{2}\int_0^{\infty}\left[\Pi^2 + (\partial_x\phi)^2\right] - \frac{1}{\pi}(\lambda - 1)(-1)^{j_0}\cos\sqrt{4\pi}\phi(x = \ell). \tag{43}$$

Since the boson sees Dirichlet boundary conditions, we have the non-trivial one-point function in the half-plane,

$$\langle e^{i\beta\phi(x)}\rangle_{\text{HP}} = \frac{1}{(2x)^{\frac{\beta^2}{4\pi}}}. \tag{44}$$

Note that the physical dimension of the r.h.s. is compatible with the values of the conformal weights $h = \bar{h} = \frac{\beta^2}{8\pi}$.

We now consider the Rényi-entropy of the interval $(0, \ell)$ using the replica approach of [6]. This involves a $p$-sheeted Riemann surface $R_p$ (with $p$ an integer) with a cut extending from the origin to the point of coordinates $(x = \ell, \tau = 0)$. We expand the partition function, and thus need for this the one point function of $e^{i\beta\phi}$ on the corresponding surface. We obtain it by uniformizing via two mappings: If $w$ is the coordinate on $R_p$, the map

$$u = -\left(\frac{w - i\ell}{w + i\ell}\right)^{1/p} \tag{45}$$

maps onto the disk $|u| \leq 1$, while the map

$$z = -i\frac{u - 1}{u + 1} \tag{46}$$

maps the disk onto the ordinary half-plane. Using Eq. (44), we obtain

$$\langle e^{i\beta\phi(w,\bar{w})}\rangle_{R_p} = \left(\frac{2\ell}{p}\right)^{2h}\frac{\left[(w - i\ell)(w + i\ell)(\bar{w} - i\ell)(\bar{w} + i\ell)\right]^{h(\frac{1}{p}-1)}}{\left[(w + i\ell)^{1/p}(\bar{w} - i\ell)^{1/p} - (w - i\ell)^{1/p}(\bar{w} + i\ell)^{1/p}\right]^{2h}}. \tag{47}$$

Here, we have used the fact that vertex operators are primary fields, and thus transform without any anomalous terms. Since two transformations are in fact required to map $R_p$ onto the half-plane $w \mapsto u \mapsto z$, we have refrained from writing the intermediate steps, and only give the final result.

The perturbative calculation of the entanglement proceeds [28] (in the Euclidian version) by considering the ratio (we set $2\frac{J'-J}{\pi} = \frac{J'-J}{\pi J} = \frac{\lambda-1}{\pi} \equiv \mu$ and take $j_0$ even for the time being),

$$R_p(\mu) \equiv \frac{Z_p}{(Z_1)^p} = \frac{\int_{\text{twist}}[\mathcal{D}\phi_1]\dots[\mathcal{D}\phi_p]\exp\left\{-\sum_{i=1}^p\left(A[\phi]_i + \mu\int d\tau_i\cos\beta\phi_i(\ell,\tau_i)\right)\right\}}{\left(\int[\mathcal{D}\phi]\exp\{-\left(A[\phi] + \mu\int d\tau\cos\beta\phi(\ell,\tau)\right)\}\right)^p}, \tag{48}$$

and the integral in the numerator is taken with the sewing conditions

$$\phi_i(0 \leq x \leq \ell, \tau = 0^+) = \phi_{i+1}(0 \leq x \leq \ell, \tau = 0^-). \tag{49}$$

---

[16]A more sophisticated approach would be to realize the cocycles using $\sigma$ matrices $\sigma_x, \sigma_y$, and diagonalize the product $\sigma_x\sigma_y = i\sigma_z$.

As usual, we trade this problem of $p$ copies of the field for a single field on $R_p$ [6]. With no modified bond, we get the known result,

$$R_p(\mu = 0) \propto \left(\frac{a}{\ell}\right)^{2h_p}, \quad h_p = \frac{c}{24}\left(p - \frac{1}{p}\right), \tag{50}$$

so

$$S = -\frac{d}{d_p}R_p\bigg|_{p=1} = \frac{1}{6}\ln\frac{\ell}{a}. \tag{51}$$

This is the usual entanglement entropy near a boundary - and *we have discarded terms of order 1 which are independent of $\mu$.*

We now proceed to calculate the ratios $R_p(\mu)/R_p(\mu = 0)$. To leading order in $|\mu| \propto 1 - \lambda \ll 1$, one finds

$$\frac{R_p(\mu)}{R_p(\mu = 0)} = 1 + \mu\left(\sum_{i=1}^{p}\int d\tau_i\langle\cos\beta\phi(\ell,\tau_i)\rangle_{R_p} - p\int d\tau\langle\cos\beta\phi(\ell,\tau)\rangle_{\text{HP}}\right). \tag{52}$$

Here, $\tau_i$ parametrizes the $p$ copies (on the $p$ sheets of $R_p$) of the line along which the perturbation is applied (in the Euclidian formulation).

It is now time to set $h = \frac{1}{2}$ for our perturbation, cf. Eqs. (42)&(44). We observe that we move from one sheet to the next by $(w - i\ell) \to e^{2i\pi}(w - i\ell)$, an operation which does not change the one-point function. We also observe that

$$\langle\cos\sqrt{4\pi}\phi(w,\bar{w})\rangle_{R_p} = \left(\frac{2\ell}{p}\right)\frac{\left[\tau^2(\tau^2 + 4\ell^2)\right]^{\frac{1}{2}\left(\frac{1}{p}-1\right)}}{\left[(\tau^2 + 4\ell^2)^{1/p} - \tau^{2/p}\right]}, \tag{53}$$

with the asymptotics $\langle\cos\sqrt{4\pi}\phi\rangle \approx (2\ell)^{-1}$, $\tau \gg \ell$, and $\langle\cos\sqrt{4\pi}\phi\rangle \approx \frac{1}{p}(2\ell)^{-1/p}\tau^{\frac{1}{p}-1}$, $\tau \ll \ell$. Because of the subtraction coming from the numerator, we see that the integral in Eq. (52) converges at large $\tau$. It also converges at small $\tau$, and can in fact be written elegantly using $\tau = 2\ell\tan\theta$, i.e.,

$$\frac{R_p(\mu)}{R_p(\mu = 0)} = 1 + 2\mu\int_0^{\frac{\pi}{2}}\frac{d\theta}{\cos^2\theta}\left[\frac{(\cos\theta)^{\frac{1}{p}-1}\cos^2\theta}{1 - (\cos\theta)^{2/p}} - p\right] \equiv 1 + 2\mu I_p. \tag{54}$$

We see from Eq. (51) that we get a correction of $\mathcal{O}(1)$ to the entanglement, given by

$$S = \frac{1}{6}\ln\left(\frac{\ell}{a}\right) + 2\mu\frac{d}{dp}I_p\bigg|_{p=1}. \tag{55}$$

Using the integral

$$\int_0^1\frac{dx}{(1 - x^2)^{3/2}}\left[1 + \frac{1 + x^2}{1 - x^2}\ln x\right] = -\frac{\pi}{6}, \tag{56}$$

we find $\frac{d}{dp}I_p\big|_{p=1} = \frac{\pi}{6}$ and thus

$$S = \frac{1}{6}\ln\left(\frac{\ell}{a}\right) + \frac{\pi}{3}\mu = \frac{1}{6}\ln\left(\frac{\ell}{a}\right) + \frac{1}{3}(\lambda - 1), \tag{57}$$

where we used that $J = \frac{1}{2}$. Recall now that this was done for $j_0$ even. The opposite term of $\mathcal{O}(1)$ is obtained for $j_0$ odd, leading to

$$\delta S = \frac{2}{3}\frac{J' - J}{J} = \frac{2}{3}(\lambda - 1). \tag{58}$$

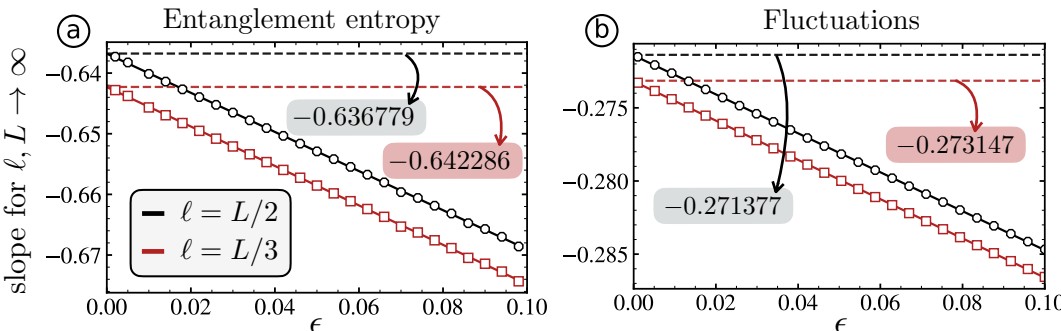

Figure 4: (a) The slope of the entanglement entropy differences $\delta S(\lambda)$ fitted in the window $\lambda = (1-\epsilon)\ldots 1$ and extrapolated to the limit $\ell, L \to \infty$ for $\ell = L/2$ (black circles) and $\ell = L/3$ (red squares). In the limit $\epsilon \to 0$, the slope is given by the analytically predicted value Eq. (59), here shown by horizontal dashed lines. (b) The same for fluctuation differences $\delta\mathcal{F}(\lambda)$, where the slope approaches the limit Eq. (82).

We see now that Eq. (24) would lead to $\delta S \approx \ln 2(\lambda - 1)$, instead of the exact result $\delta S = \frac{2}{3}(\lambda - 1)$, hence establishing that the ellipse approximation cannot be exact.

We note that finite size effects can be studied in perturbation theory as well. Following again [6], we find that the leading term is obtained upon using the more general integral

$$\frac{d}{dp} \int_0^\infty dx \frac{1}{\sqrt{1 + \sin^2 \frac{\pi \ell}{L} x^2}} \left[ \frac{\left(x^2(1+x^2)\right)^{\frac{1}{2}\left(\frac{1}{p}-1\right)}}{(1+x^2)^{\frac{1}{p}} - x^{\frac{2}{p}}} - p \right], \tag{59}$$

which is equal to $\frac{\pi}{6}$ for $\ell \ll L$, and $\approx 0.500125$ for $\ell = \frac{L}{2}$. We then find

$$\delta S = 0.636779(\lambda - 1), \tag{60}$$

instead of the $\frac{2}{3}$ slope for an infinite system. When instead fixing $\ell = \frac{L}{3}$, the slope turns out to be given by $0.642286$. We can check these results by 'ab-initio' measurements of the slopes close to $\lambda \lesssim 1$. Using a varying fitting window of $\lambda = (1-\epsilon)\ldots 1$ and extrapolating the slopes to infinite system size, we see how the slope close to $\lambda \lesssim 1$ approaches the analytically predicted value Eq. (59) for vanishing window size $\epsilon \to 0$, cf. Fig. 4 (a). The growing discrepancies for larger fitting values away from $\epsilon \gtrsim 1$ suggests that there are other terms, possibly of the form $\lambda \log \lambda$, which we have not considered in our lowest order perturbative calculation.

To the order we have considered, the fact that the entanglement gets finite terms of $\mathcal{O}(1)$ arises technically from the facts

(i) that the term coupled to $\lambda - 1$ acquires a finite expectation value in the presence of a boundary, and

(ii) that, while this expectation value decays with $\ell$, the integral over imaginary time leads to a finite contribution.

While calculating entanglement perturbatively at vanishing temperature leads usually to difficulties [28], we note that no divergence is encountered at leading order. At this order, the (anti-)symmetry under $\lambda \to \frac{1}{\lambda}$ also appears clearly.

While our calculation started out with a specific model (the XX chain with one modified bond), any other system in the universality class of free fermions and with a set of modified bonds should be equivalent to the field theory action of a free boson perturbed by the term

Eq. (42) up to *irrelevant* terms. The same calculation should then apply, and we expect therefore our term of $\mathcal{O}(1)$ to be a universal function of the effective amplitude in the Hamiltonian Eq. (43).

## 3.2 Charge fluctuations

We first recall briefly what happens without an impurity or a boundary. Using the same factorization Eq. (38) as before, we have for the charge density

$$c_j^\dagger c_j \mapsto \psi_R^\dagger \psi_R + \psi_L^\dagger \psi_L + e^{2iK_F j}\psi_R^\dagger \psi_L + e^{-2iK_F j}\psi_L^\dagger \psi_R \,, \tag{61}$$

where, at half-filling, $K_F = \frac{\pi}{2}$. One usually considers (but see below) that the oscillating terms can be neglected when considering the fluctuations of the charge $Q = \sum_j c_j^\dagger c_j$ in a given interval. Going over to a continuous description $\rho(x = ja) = c_j^\dagger c_j$, we obtain

$$\langle \rho(x)\rho(x')\rangle_c \approx \langle : \psi_R^\dagger \psi_R : (x) : \psi_R^\dagger \psi_R : (x')\rangle + R \leftrightarrow L + \text{rapidly oscillating terms} \,. \tag{62}$$

We now recall that

$$\begin{aligned}\langle \psi_R^\dagger(x)\psi_R(x')\rangle &= \frac{i}{2\pi(x-x')} \,, \\ \langle \psi_L^\dagger(x)\psi_L(x')\rangle &= -\frac{i}{2\pi(x-x')} \,,\end{aligned} \tag{63}$$

so we find, with $Q = \int_0^\ell \rho(x)dx$,

$$\langle (Q - \langle Q\rangle)^2\rangle = 2 \times -\frac{1}{4\pi^2}\int_0^\ell dx dx' \frac{1}{(x-x')^2} = \frac{1}{\pi^2}\ln\frac{\ell}{a} \,, \tag{64}$$

where we regularized UV divergences with the cut-off $a$[17]. Note that this result is pretty much the bosonic propagator since, upon bosonizing and by using $\phi = \phi_R + \phi_L$, we obtain

$$\begin{aligned}\langle \rho(x)\rho(x')\rangle_c &\approx \frac{1}{\pi}\left(\langle \partial_x \phi_R(x)\partial_x \phi_R(x')\rangle + \langle \partial_x \phi_L(x)\partial_x \phi_L(x')\rangle\right) + \ldots \\ &\quad + \text{rapidly oscillating terms} \\ &\approx \frac{1}{\pi}\langle \partial_x \phi(x)\partial_x \phi(x')\rangle \,,\end{aligned} \tag{65}$$

and therefore

$$\langle (Q - \langle Q\rangle)^2\rangle \equiv \mathcal{F}_A \approx \frac{1}{\pi}\langle [\phi(\ell) - \phi(0)]^2\rangle \,. \tag{66}$$

Using the free boson propagator on the infinite line,

$$\langle \phi(x)\phi(x')\rangle_c = -\frac{1}{2\pi}\ln\frac{|x-x'|}{a} \,, \tag{67}$$

we recover Eq. (35).

We now turn to the continuous description of the impurity in the fermionic language. We have, to leading order,

$$c_j^\dagger c_{j+1} + \text{h.c.} \mapsto 2i(-1)^j(\psi_L^\dagger \psi_R - \psi_R^\dagger \psi_L) \,. \tag{68}$$

---

[17]Although we use the same notation, this cutoff is only proportional to the cutoff discussed in the numerical calculations.

Solving the Schrödinger equation in the continuum limit gives, for a modified bond at position $j_0$ and setting $\ell \equiv j_0 a$,

$$\begin{pmatrix} \psi_R(\ell+) \\ \psi_L(\ell-) \end{pmatrix} = \begin{pmatrix} \cos\xi & \sin\xi \\ -\sin\xi & \cos\xi \end{pmatrix} \begin{pmatrix} \psi_R(\ell-) \\ \psi_L(\ell+) \end{pmatrix}. \tag{69}$$

This defines the phase shift $\xi$ at the Fermi-surface. It is, in general, a non-universal quantity which must be calculated starting from the lattice model. In our case, we find

$$\xi = (-1)^{j_0} \left[ \frac{\pi}{2} - 2\arctan\frac{J'}{J} \right], \tag{70}$$

where the oscillating factor $(-1)^{j_0}$ originates from the alternating term in Eq. (68). We see that shifting the weak bond by one site (i.e., exchanging the odd and even cases) amounts to changing the sign of $\xi$, or equivalently changing $\lambda = \frac{J'}{J} \to \frac{1}{\lambda} = \frac{J}{J'}$.

In the presence of a weak link *but no boundary*, we can now see how this result is modified. We can for instance decompose the fermion fields into modes. Assuming the weak link is at $x = \ell$, this gives

$$\left.\begin{aligned} \psi_R(x,t) &= \int_{k>0} \frac{dk}{2\pi} \left[ e^{ik(x-t)}\alpha(k) + e^{-ik(x-t)}\beta^\dagger(k) \right] \\ \psi_L(x,t) &= \int_{k>0} \frac{dk}{2\pi} \left[ e^{-ik(x+t)}\bar{\alpha}(k) + e^{ik(x+t)}\bar{\beta}^\dagger(k) \right] \end{aligned}\right\} \quad \text{for} \quad x < \ell, \tag{71}$$

and similarly

$$\left.\begin{aligned} \psi_R(x,t) &= \int_{k>0} \frac{dk}{2\pi} \left[ e^{ik(x-t)}a(k) + e^{-ik(x-t)}b^\dagger(k) \right] \\ \psi_L(x,t) &= \int_{k>0} \frac{dk}{2\pi} \left[ e^{-ik(x+t)}\bar{a}(k) + e^{ik(x+t)}\bar{b}^\dagger(k) \right] \end{aligned}\right\} \quad \text{for} \quad x > \ell, \tag{72}$$

with matching conditions

$$\begin{aligned} a &= \cos\xi\,\alpha + e^{-2ikL}\sin\xi\,\bar{a}, \\ b^\dagger &= \cos\xi\,\beta^\dagger + e^{2ikL}\sin\xi\,\bar{b}^\dagger, \\ \bar{a} &= \cos\xi\,\bar{a} - e^{2ikL}\sin\xi\,\alpha, \\ \bar{\beta}^\dagger &= \cos\xi\,\bar{b}^\dagger - e^{-2ikL}\sin\xi\,\beta^\dagger. \end{aligned} \tag{73}$$

We must now consider the modes $\alpha, \beta^\dagger, \bar{a}, \bar{b}^\dagger$ (and their conjugates) as independent, while $\bar{\alpha}, \bar{\beta}^\dagger, a, b^\dagger$ (and their conjugates) follow from the matching relations. While the RR and LL correlators are not modified, we now have a non-zero RL correlator,

$$\langle \psi_R^\dagger(x)\psi_L(x') \rangle = -\frac{\sin\xi}{(2i\pi)(x+x'-2\ell)}. \tag{74}$$

This leads to fluctuations (for an interval in the bulk, with an impurity on one of its borders) of

$$\mathcal{F}_{A,\text{imp}} \approx \frac{1}{2\pi^2}(1+\cos^2\xi)\ln\frac{\ell}{a}. \tag{75}$$

We see that the slope interpolates between $\frac{1}{\pi^2}$ when $\xi = 0$ (no impurity) to $\frac{1}{2\pi^2}$ when the chain is cut in half. The 'impurity part' has a slope proportional to $\cos^2\xi = s^2 = \frac{4(JJ')^2}{(J^2+J'^2)^2}$, where we now recover the result Eq. (29) and see how $C$ plays a role similar to the effective central charge.

The case with a boundary and no impurity is slightly more convenient to handle via bosonization. In this case, the left and right components of the bosonic field are not independent any longer due to the Dirichlet boundary conditions,

$$\langle \phi(x,\tau)\phi(x',\tau') \rangle = -\frac{1}{4\pi}\left( \ln|z-z'|^2 - \ln|z-\bar{z}'|^2 \right), \tag{76}$$

where $z \equiv -\tau + ix$. It follows that (using $\phi(0) = 0$),

$$\langle [\phi(0) - \phi(\ell)]^2 \rangle = \langle [\phi(\ell)]^2 \rangle = \frac{1}{2\pi} \ln \frac{\ell}{a} + \frac{1}{2\pi} \ln 2, \qquad (77)$$

and thus

$$\mathcal{F}_{A,\text{bdr}} = \frac{1}{2\pi^2} \ln \frac{2\ell}{a} . \qquad (78)$$

While the field theory does not control the terms of order one, it does control their difference, as seen by comparing Eqs. (64),(78) with Eqs. (27),(32).

Finally, when both the boundary and the impurity are present, we can do perturbation theory, once again in the bosonized language. What matters then are the connected terms in the correlator,

$$\langle [\phi(\ell, \tau = 0)]^2 \cos \beta \phi(\ell, \tau) \rangle_c = \frac{1}{4\pi} \left[ \ln \left( 1 + \frac{4\ell^2}{\tau^2} \right) \right]^2 \times \frac{1}{2L} . \qquad (79)$$

Hence, the leading correction to the charge fluctuations is given by

$$\begin{aligned}
\langle (Q - \langle Q \rangle)^2 \rangle^{(1)} &= \mu \times \frac{1}{\pi} \times \frac{1}{4\pi} \times \frac{1}{2\ell} \int_{-\infty}^{\infty} d\tau \left[ \ln \left( 1 + \frac{4\ell^2}{\tau^2} \right) \right]^2 \\
&= \frac{\mu}{2\pi^2} \int_0^{\infty} dx \left[ \ln \left( 1 + \frac{1}{x^2} \right) \right]^2,
\end{aligned} \qquad (80)$$

where the integral is tabulated and yields $4\pi \ln 2$. Subtracting the opposite term for the odd case, it follows that

$$\langle (Q - \langle Q \rangle)^2 \rangle_e^{(1)} - \langle (Q - \langle Q \rangle)^2 \rangle_o^{(1)} = \frac{4\mu}{\pi} \ln 2 = \frac{4}{\pi^2} \frac{J' - J}{J} \ln 2 = \frac{4 \ln 2}{\pi^2} (\lambda - 1), \qquad (81)$$

where we recall that $\frac{J'}{J} = \lambda$. With $\frac{4 \ln 2}{\pi^2} \approx 0.280922 \neq \frac{1}{4}$ we see that the result is, again, not supported by the naive ellipse shape mentioned previously. We note that the leading term $\propto \ln \ell$ does not get corrections to first order in $\mu$, which is in agreement with the formula for the slope in Eq. (75).

The foregoing calculation holds when the interval from the edge to the impurity is part of a half-infinite system. We can otherwise expect finite-size corrections, like for the entanglement. It is possible to calculate the effect of these corrections on the first order term by evaluating the propagators on a strip instead of in the half-plane, using a conformal map, which yields

$$\langle (Q - \langle Q \rangle)^2 \rangle^{(1)} = \frac{\mu}{2\pi^2} \int_0^{\infty} \frac{dx}{\sqrt{1 + \sin^2 \frac{\pi \ell}{L} x^2}} \left[ \ln \left( 1 + \frac{1}{x^2} \right) \right]^2 . \qquad (82)$$

We find that $\frac{4 \ln 2}{\pi^2} \approx 0.280922$ is replaced by $0.271377$ when $\ell = \frac{L}{2}$, or by $0.273147$ when $\ell = \frac{L}{3}$ - these are small but non negligible effects. We again check these results by 'ab-initio' measurements of the slopes close to $\lambda \lesssim 1$, obtained by diagonalizing the tight binding Hamiltonian. As for the entanglement entropy, we observe that the slope is given by Eq. (82) for vanishing fitting window, cf. Fig. 4 ⓑ.

Finally, we note that the behavior of the slope of $\delta \mathcal{F}$ close to $\lambda = 1$ can also be derived from first order perturbation theory around the exact plane wave solution of the homogeneous chain with open boundaries, yielding identical results - see the Appendix for a discussion.

# 4 Extended impurities (towards the SSH model)

In this section, we would like to illustrate the fact that the entanglement and fluctuation oscillations only depend on the phase shift at the Fermi level. Let us for this purpose consider a system with $N_{\text{imp}}$ impurity bonds arranged in an alternating fashion, i.e.,

$$
\begin{aligned}
H = -J \sum_m \big( c^\dagger_{m+1} c_m + \text{h.c.} \big) - J \big[ (\lambda^{\langle 0,1 \rangle} - 1) c^\dagger_{j_0} c_{j_0+1} + (\lambda^{\langle 2,3 \rangle} - 1) c^\dagger_{j_0+2} c_{j_0+3} + \dots \\
+ (\lambda^{\langle 2N_{\text{imp}}, 2N_{\text{imp}}+1 \rangle} - 1) c^\dagger_{j_0+2N_{\text{imp}}} c_{j_0+2N_{\text{imp}}+1} + \text{h.c.} \big],
\end{aligned} \tag{83}
$$

where $\lambda^{\langle i,j \rangle}$ is the impurity strength between sites $j_0 + i, j_0 + j$.

## 4.1 General results

To solve the system, we make a plane wave ansatz,

$$
|k\rangle = \sum_{j=-\infty}^{j_0-1} \big( A_k e^{ikj} + B_k e^{-ikj} \big) |j\rangle + \sum_{j=j_0}^{j_0+2N_{\text{imp}}-1} c_k^j |j\rangle + \sum_{j=j_0+2N_{\text{imp}}}^{\infty} \big( C_k e^{ikj} + D_k e^{-ikj} \big) |j\rangle, \tag{84}
$$

with

$$
H|k\rangle = -2J \cos(k) |k\rangle. \tag{85}
$$

By comparing coefficients after applying the Hamiltonian to Eq. (84), we find for $c_k^0$ and $c_k^1$

$$
\begin{aligned}
c_k^0 &= A_k + B_k, \\
\lambda^{\langle 0,1 \rangle} c_k^1 &= A_k e^{ik} + B_k e^{-ik}.
\end{aligned} \tag{86}
$$

The coefficients $c_k^3 \dots c_k^{2N_{\text{imp}}-1}$ are then constructed with the recursion relation

$$
c_k^n = \begin{cases} 2\cos(k) c_k^{n-1} - \lambda^{\langle n-2, n-1 \rangle} c_k^{n-2}, & \text{for } n \text{ even}, \\ \frac{1}{\lambda^{\langle n-1, n \rangle}} \big( 2\cos(k) c_k^{n-1} - c_k^{n-2} \big), & \text{for } n \text{ odd}. \end{cases} \tag{87}
$$

This now expresses all tight-binding coefficients lying inside the impurity with the plane wave coefficients $A_k, B_k$. We further get two additional equations relating $c_{2N_{\text{imp}}-1}$ and $c_{2N_{\text{imp}}-2}$ with $C_k$ and $D_k$, given by

$$
\begin{aligned}
2\cos(k) c_k^{2N_{\text{imp}}-1} &= \lambda^{\langle 2N_{\text{imp}}-2, 2N_{\text{imp}}-1 \rangle} c_k^{2N_{\text{imp}}-2} + C_k e^{2ikN_{\text{imp}}} + D_k e^{-2ikN_{\text{imp}}}, \\
c_k^{2N_{\text{imp}}-1} &= C_k e^{ik(2N_{\text{imp}}-1)} + D_k e^{-ik(2N_{\text{imp}}-1)}.
\end{aligned} \tag{88}
$$

Eqs. (89),(87) fully solve the $N_{\text{imp}}$ alternating impurity system. In case of half-filling, these equations simplify considerably, such that

$$
c_k^n = \begin{cases} (-1)^{n/2} (A_k + B_k) \prod_{i=0}^{n/2-1} \lambda^{\langle 2i, 2i+1 \rangle}, & \text{for } n \text{ even}, \\ (-1)^{(n+1)/2} i (A_k - B_k) / \prod_{i=0}^{(n-1)/2} \lambda^{\langle 2i, 2i+1 \rangle}, & \text{for } n \text{ odd}. \end{cases} \tag{89}
$$

Now, using Eq. (88), we find

$$
\begin{aligned}
(A_k + B_k) \prod_{i=0}^{N_{\text{imp}}-1} \lambda^{\langle 2i, 2i+1 \rangle} &= C_k + D_k, \\
A_k - B_k &= (C_k - D_k) \prod_{i=0}^{N_{\text{imp}}-1} \lambda^{\langle 2i, 2i+1 \rangle},
\end{aligned} \tag{90}
$$

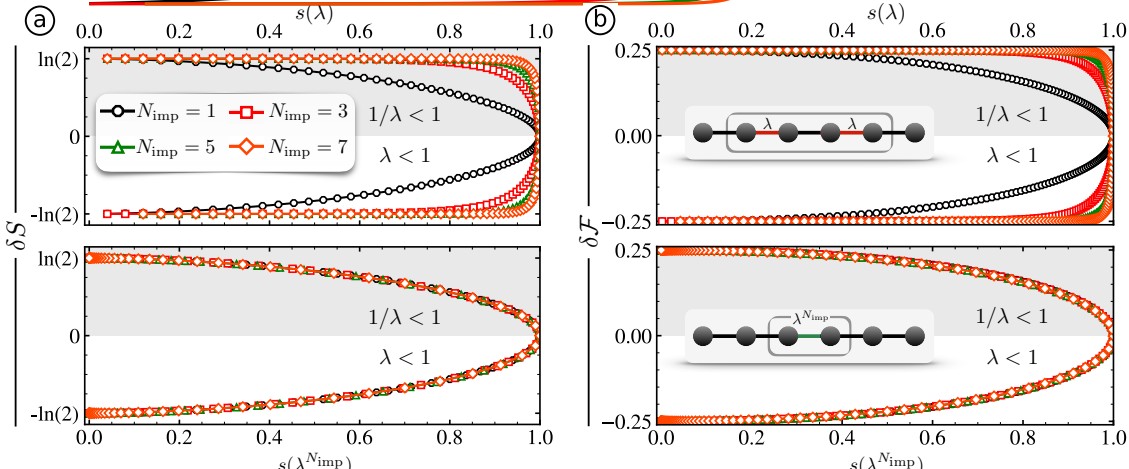

Figure 5: Parity effects of ⓐ the entanglement entropy and ⓑ the fluctuations when transitioning from the single impurity link to an "SSH-like" impurity with a collection of alternating bonds $J, \lambda J$. Upon increasing the number of impurities $N_{\text{imp}} = 1, 3, 5, 7$ with all $\lambda^{\langle 2n, 2n+1 \rangle} = \lambda$, the differences quickly saturate to $\pm \ln 2$ (upper panel ⓐ) and $\pm 1/4$ (upper panel ⓑ). This can be understood in terms of an effective, stronger single impurity with strength $\lambda^{N_{\text{imp}}}$. When rescaling $\delta S$ and $\delta \mathcal{F}$ accordingly, the curves become equivalent (lower panels).

where $i = 0..N_{\text{imp}} - 1$ now runs through all impurity bonds.

Finally, with $\prod_i \lambda^{\langle 2i, 2i+1 \rangle} = \Lambda$, this results in

$$\begin{pmatrix} B \\ C \end{pmatrix} = \begin{pmatrix} \frac{2\Lambda^2}{1+\Lambda^2} & \frac{1-\Lambda^2}{1+\Lambda^2} \\ -\frac{1-\Lambda^2}{1+\Lambda^2} & \frac{2\Lambda^2}{1+\Lambda^2} \end{pmatrix} \begin{pmatrix} D \\ A \end{pmatrix}. \tag{91}$$

We thus see that the collection of impurities behaves, in fact, like a single one of effective strength $\Lambda$. Comparing with Eqs. (69),(70), we see that the phase shift at the Fermi surface[18] is now given by

$$\xi = (-1)^{j_0} \left[ \frac{\pi}{2} - 2 \arctan \left( \prod_i \lambda^{\langle 2i, 2i+1 \rangle} \right) \right]. \tag{92}$$

On the other hand, we can study numerically the fluctuations or the entanglement for these extended impurities as well. In analogy with the case of the single modified bond ($N_{\text{imp}} = 1$), we define the entanglement and fluctuation differences as the difference between even and odd parities of the part of the system situated between the boundary and the border of the extended impurity. In practice, this corresponds to comparing two systems where the cut defining the subsystem as well as the impurity are both shifted by one site.

Restricting to the simplest case where all impurity strengths in the chain are identical, $\Lambda = \lambda^{N_{\text{imp}}}$, we find for instance that the curves for the $N_{\text{imp}}$-impurity system coincide with those for a single impurity upon the rescaling $\lambda \to \lambda^{N_{\text{imp}}}$, see Fig. 5. Here, we always choose an odd number of impurities and cut the system through the central impurity bond. The observation that the fluctuations become identical upon rescaling $\lambda \to \lambda^{N_{\text{imp}}}$ is a clear indication that the parity effects in the scaling limit are governed by the physics at the Fermi level, ultimately supporting our conjecture that the differences are universal.

---

[18]The phase shift depends on $k$ in general. However, it becomes independent of $k$ (and equal to $\xi$) for low-energy excitations, that is excitations whose energy is much smaller than the band-width, and this irrespective of the modified couplings.

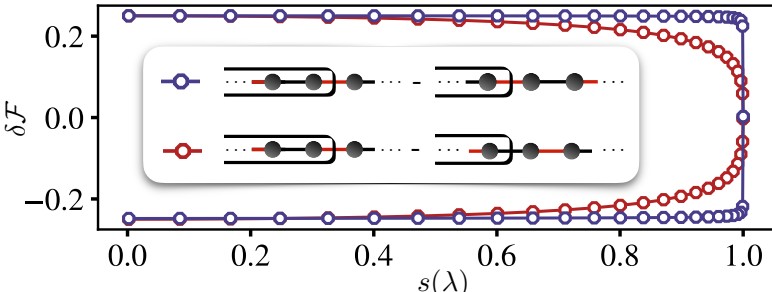

Figure 6: Comparing two different ways of defining the fluctuation differences between the parities. (i) We both change the subsystem's parity as well as the topological phase, i.e., in the language of impurities, we "move" the impurity together with the subsystem's border (see the blue curve). Though locally the systems look the same at the subsystem's border, the edge modes appearing in the topologically non-trivial phase lead to a 1/4 contribution for all $\lambda$. (ii) While changing the parity of the subsystem, the geometry of the underlying model is fixed. Here, only local differences are seen, leading to a functional dependence different from the step function.

Note that these results are indeed independent on how one defines the subsystem's border. For instance, when defining the cut to go through one of the outermost impurity bonds instead of the central one, we numerically checked that the curves are identical[19]- which is expected since the phase shift at the Fermi surface is independent of the definition of the subsystem.

When again focusing on the upper curves in Fig. 5, we see that $\delta S(\lambda)$ $(\delta\mathcal{F}(\lambda))$ approaches a step function taking values $\pm\ln 2$ $(\pm 1/4)$ for $\lambda \gtrsim 1$ in the limit of an infinitely large $N_{\text{imp}} \gg 1$ impurity.

## 4.2 SSH model

We now observe that entanglement entropy differences in the SSH model [29,30] look similar to very large alternating impurities. It is interesting to see what becomes of our observations in the former case. First, we note that the way we define the entanglement entropy and the fluctuation differences both for the single impurity bond as well as the extended (alternating) impurities corresponds to comparing the two distinct topological phases in the limit of a pure SSH model (all weak and strong bonds are exchanged and the border of the subsystem is moved by one unit).

In this case, the local environment around the border of the subsystem looks identical for both parities, and its influence gets cancelled when taking the difference, see the upper part of the inset of Fig. 6. However, the boundary of the system features localized modes in the topologically non-trivial regime, which remain when we subtract the entanglement from the trivial regime (where the edge modes don't exist). This, in turn, results in an entanglement entropy (fluctuation) difference of exactly $\ln 2$ (1/4) - in fact for all values of $\lambda$, cf. the blue curve in Fig. 6.

Had we, on the other hand, fixed the geometry of the SSH chain and focused on entanglement and fluctuation differences arising from merely shifting the subsystem's border from even $\leftrightarrow$ odd, we would only observe local differences around the impurity (cf. the lower part of the inset of Fig. 6). This results in $\ln 2$ (1/4) differences of the entanglement (fluctuations) only in the limit $\lambda = 0$, in which case we can again rely on the valence bond picture: *local*

---

[19]In this case both definitions have the same $N_{\text{imp}}$ limit and both scale as described above, making the curves indeed identical.

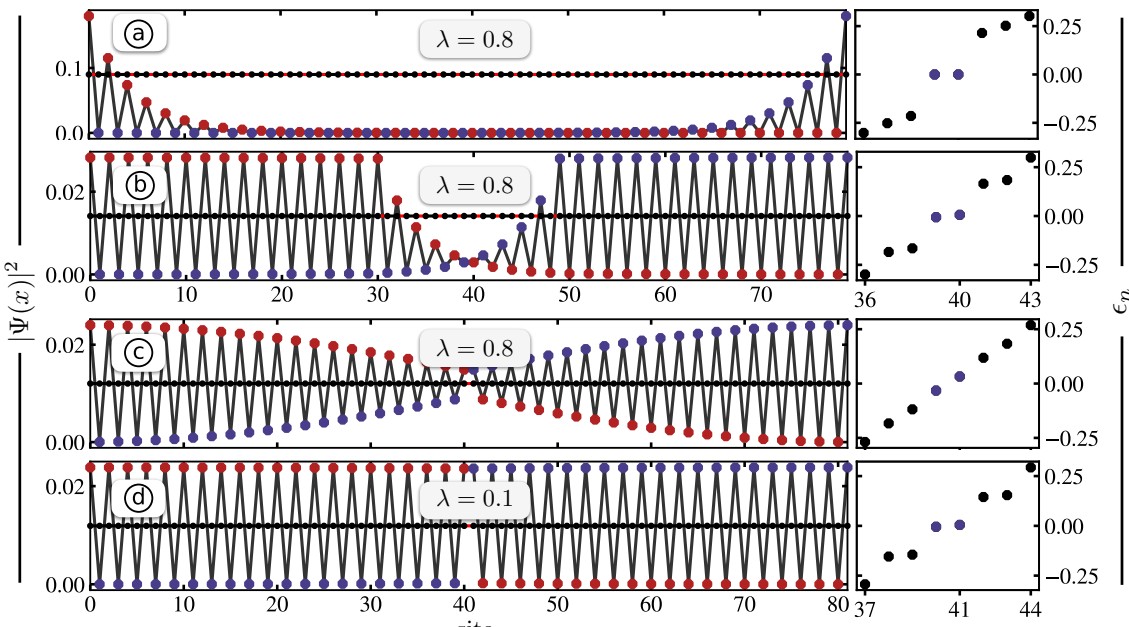

Figure 7: Modes $|\Psi(x)|^2$ in vicinity of zero energy for various tight-binding models. Values of $|\Psi(x)|^2$ are marked in an alternating fashion by red and blue dots for A and B sublattices. Each model is illustrated in the center of the left panel, with strong (weak) bonds marked by black (red) lines. Right panels show the energy spectra close to zero energy, with the two symmetric modes closest to $\epsilon = 0$ marked in blue. ⓐ The pure SSH chain features localized zero energy modes, having support only on sublattice A (B) for the left (right) edge mode. Here, $\lambda = 0.8$. ⓑ An SSH impurity of length 20 attached to two leads of size 30 on either side, again with $\lambda = 0.8$. The long SSH impurity polarizes the wave functions close to zero energy, having support only on the A (B) sublattice on the left (right) to the extended impurity. ⓒ A single impurity bond with 40 metallic sites to each left and right side for $\lambda = 0.8$. ⓓ The same as ⓒ with $\lambda = 0.1$.

valence bonds between dimers lead to differences of $\ln 2$ $(1/4)$ when changing the parity of the subsystem's border, see the red curve in Fig. 6.

It is not clear *a priori* to what extent the limit $N_{\mathrm{imp}} \gg 1$ can be considered as identical with a pure SSH model since, in all our calculations, we have always considered an impurity embedded in a gapless bulk (or metallic) system [31]. We observe however that the corresponding curves for the entanglement or fluctuation differences are the same step functions. This means that the two leads coupling to the impurity become effectively decoupled in the limiting case of spatially large SSH-type impurities, ultimately also leading to a contribution of $\ln 2$ $(1/4)$ to the entanglement (fluctuation) differences for all impurity strengths. It seems therefore that the phenomenology of the SSH model and the SSH-type impurity are similar: in the former case the differences arise from the edge modes appearing in the non-trivial phase, and in the latter they arise from the leads being effectively decoupled[20].

This decoupling of the leads is further confirmed by an analysis of the single particle energies and wave functions of the SSH model when we add leads of increasing length to the edges starting with either of the two topological phases. We observe that when we add leads to the edges of a non-trivial SSH chain, the zero energy modes remain- though the gap around them

---

[20]In the language of the SSH model, this corresponds to having negligible overlap between the two exponentially decaying edge modes.

closes. Nevertheless, the central SSH chain fully polarizes the system in the same sense it does for the conventional (full) SSH chain, i.e., the wave function of the zero modes localized on the right (left) edge only have support on the A (B) sublattice.

This is illustrated in Fig. 7. The left panel in ⓐ shows a linear combination of the zero modes to have weight on both boundaries in the non-trivial SSH chain, localized at the edges of the system on sublattice A (red dots) on the left and B (blue dots) on the right. The single particle spectrum around zero energy is shown in the right panel of Fig. 7 ⓐ, whereby edge (bulk) modes are marked by blue (black) dots. When coupling metallic leads to an SSH chain of size 20, we see that - up to a small split due to the shorter SSH part- (approximate) zero modes are still present, and the corresponding wave function is polarized on sublattice A (B) to the left (right) of the chain, see Fig. 7 ⓑ. When now keeping the same system size but further shrinking the SSH impurity part, we notice how the polarization around the impurity weakens but qualitatively remains intact even in the extreme limit of a single impurity bond, shown for $\lambda = 0.8$ in Fig. 7 ⓒ. By decreasing the ratio $\lambda$, the polarization is enhanced, shown in Fig. 7 ⓓ for $\lambda = 0.1$. Indeed, we find that for $\lambda < 1$ the energy split of the two modes around zero energy scales as $\sim e^{\ln(\lambda)N_{\text{imp}}} = \lambda^{N_{\text{imp}}}$, akin to the scaling law we found for the fluctuation and entanglement differences[21].

In the language of the earlier impurity discussion, the above examples shown in Fig. 7 correspond to the "even case". If we instead consider the trivial phase of the SSH chain ("odd case") and couple leads to it, there are no wave functions that are strongly polarized on one sublattice only. Hence, when comparing the two opposing geometries (i.e. "even" and "odd", which corresponds to non-trivial and trivial in the full SSH limit), we can get an intuition for the resemblance of terms of $\mathcal{O}(1)$ of the SSH and single impurity bond model: both systems feature low energy modes that polarize the system on a scale that grows exponentially with the length of the (central) SSH chain $\propto \lambda^{N_{\text{imp}}}$. This, in turn, leads to non-quantized values of entanglement/fluctuation differences for small impurity lengths and strengths.

Note that the problem we considered in this paper is fundamentally different from the SSH model, in particular because the theory in the bulk is conformal instead of being gapped. However, it is tempting to think of the oscillations we have exhibited as some sort of measure of "topological phase precursor", even though the meaning of the (non-topological) numbers $\delta S$ ($\delta \mathcal{F}$) other than $\ln 2$ ($1/4$) remains unclear.

Lastly, note that the above scaling results would break down if the modified bonds did not obey the alternating pattern where unmodified bonds sit between two modified ones. An interesting example with such a different behavior is the case where two successive bonds are modified, leading to the so-called resonant level model which we discuss now.

## 5 The quantum dot and RG flows

We consider the case of a "quantum dot", which is realized by considering a model with two successive modified bonds, and Hamiltonian

$$H = -J \sum_{\substack{j=1 \\ j \neq j_0, j_0+1}}^{\infty} c_{j+1}^{\dagger} c_j + c_j^{\dagger} c_{j+1} - J' \left( c_{j_0+1}^{\dagger} c_{j_0} + c_{j_0}^{\dagger} c_{j_0+1} + c_{j_0+2}^{\dagger} c_{j_0+1} + c_{j_0+1}^{\dagger} c_{j_0+2} \right). \tag{93}$$

It is customary in this case to think of the site $j_0 + 1$ as special (the "dot"), and denote $c_{j_0+1}^{(\dagger)} = d^{(\dagger)}$. In this form, we are dealing with the resonant level model near a boundary.

The behavior of this model is profoundly different from the physics of the model with a single modified link [27]. This can be understood if one observes that (in the absence of a

---

[21]In both cases, a consequence of the scaling of the phase shift at the Fermi energy.

boundary), the latter has the symmetry of reflection through a bond, while the former has symmetry of reflection through a site. For the single modified link, the term $J'-J$ corresponds to a marginal perturbation, with energy independent scattering in the low-energy limit. In contrast, the case with two successive modified links exhibits the physics of "healing". When $J' \approx J$ (and the chain is almost homogeneous), the impurity behaves as an irrelevant perturbation of the unmodified chain, and the system still looks homogeneous at low-energy (large scales). When $J'$ is small, the tunneling term $J'$ is a relevant perturbation of the two-decoupled chains limit, and the system flows again to a homogeneous one at at low-energy. For the XX case, the dimension of the tunneling term coupled to $J'$ in the limit $J' \to 0$ is 1/2 (it is 2 in the limit $J' \approx J$, while it is equal to the marginal value 1 for the single modified link). The ensuing physics is close to the one of the Kondo model in the spin channel [32].

The presence of an RG flow means that there is a characteristic crossover scale which we shall denote $T_B \propto (J')^2 \propto \lambda^2$ (like before, we set $J' = \lambda J$). The dependency of physical quantities on $\ell$ and $T_B$ will depend on their behaviour under RG.

While the foregoing considerations apply even to an interacting system (of XXZ type), they can be made more explicit in the simple XX case. In this case indeed, the transmission probability (see e.g. [33]) for one-particle energies $\epsilon \ll J$ reads

$$|t|^2 \approx \frac{1}{1 + \frac{J^2}{4}\left(\frac{\epsilon}{J'^2}\right)^2} \, , \tag{94}$$

instead of being a constant ($= \cos^2 \xi$) a low-energy like in the case of a single modified link. This means that, if we consider the limit $J' \to 0$ while sending the characteristic energy scale (e.g. the temperature, or, in our case, $\ell^{-1}$) also to zero, some non-trivial scaling behavior of physical properties emerges[22]

The scaling form of the entanglement or fluctuations is quite involved [22], and it is simpler to consider derivatives. As discussed in the introduction already, we observe that these derivatives exhibit different, finite terms of $\mathcal{O}(1)$[23] depending on the parity of $\ell$. Setting

$$\begin{aligned}
\frac{dS^e_{A,\text{imp+bdr}}}{d\ln\ell} &= F(\ell T_B) + f^e(\ell T_B) \, , \\
\frac{dS^o_{A,\text{imp+bdr}}}{d\ln\ell} &= F(\ell T_B) + f^o(\ell T_B) \, ,
\end{aligned} \tag{95}$$

we consider in particular the difference of terms of $\mathcal{O}(1)$, given by

$$f^e(\ell T_B) - f^o(\ell T_B) \equiv \delta \frac{dS(\ell T_B)}{d\ln\ell} \, . \tag{96}$$

We follow the same strategy as previously for the single impurity bond, Sec. 2.1 & 2.2, as well as the alternating impurity geometry, Sec. 4, and compute entanglement entropy and fluctuations for varying system sizes while fixing the cut through the system, which we here choose to be the left impurity bond (i.e. excluding the quantum dot in region $A$, see the inset of Fig. 8 ⓐ). By again considering a fixed ratio $Z = 2$, we compute the difference of terms of $\mathcal{O}(1)$, Eq. (96). We observe that it exhibits a particularly interesting, non-monotonic behavior with a maximum close to the crossover length scale $L \approx \lambda^{-2}$, see the lower panel of Fig. 8 ⓐ. In contrast with the entropies themselves, the difference between the even and odd cases has a simple scaling form, and can be seen to interpolate smoothly between the UV ($\ell T_B \ll 1$) and

---

[22]In contrast, at fixed $J'$ and low enough energy, the system appears healed and the impurity has no effect.

[23]The terms $f^{e,o}$ below originate from terms of $\mathcal{O}(1)$ in the entanglement indeed. Notice however that when taking derivatives, these terms become in fact of the same order at the leading one. Since we focus on differences between even and odd however, this does not matter.

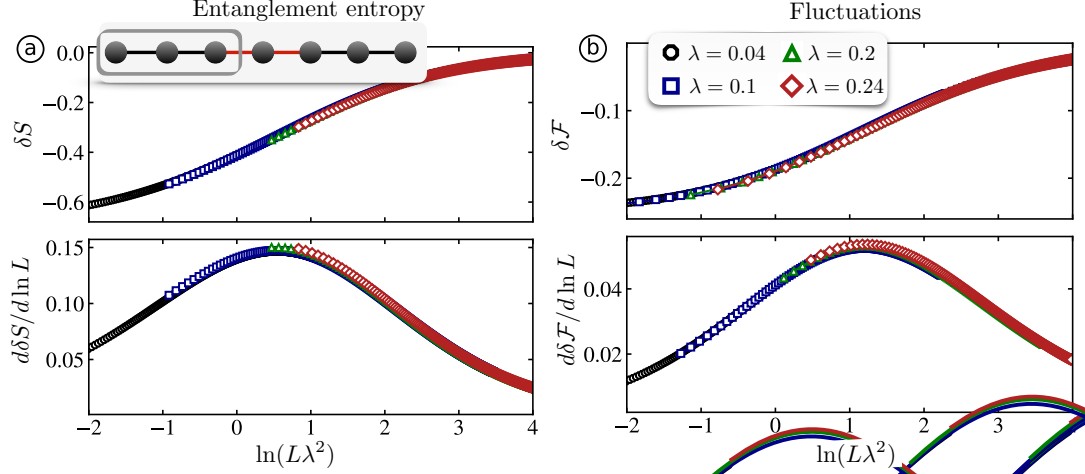

Figure 8: ⓐ Entanglement entropy for varying positions of the quantum dot, excluding the dot from subsystem $A$, as illustrated in the inset of the upper panel. As for the single impurity bond, $\delta S$ is defined as the difference of even and odd parities of the quantum dot position. The derivative of the difference with respect to $\ln L$, where L is the total system size, is shown in the lower panel, where a crossover can be observed at $L\lambda^2 \sim 1$. ⓑ The same for charge fluctuations.

the IR ($\ell T_B \gg 1$) regimes (see the upper panel in Fig. 8 ⓐ). Once again, very similar results are obtained for the fluctuations, as shown in Fig. 8 ⓑ. Note that we find similar results for larger ratios $Z = L/\ell$, however with considerably more noticeable finite size effects, which is why we fix the quantum dot to lie in the center of the chain, $Z = 2$, in our considerations.

We further note that it is possible to carry out perturbative calculations in this case as well, but the results are not particularly useful and we refrain from presenting them here.

## 6 Conclusions

In conclusion, we have uncovered parity effects in entanglement and fluctuations that do not decay with distance when the boundary is combined with an impurity in an XX chain. These parity effects contain interesting physical information: they can be construed as indicating the presence of a topological phase in the SSH model for impurities preserving the symmetry of reflection through a link (conformal defects), and exhibit Kondo type physics for impurities preserving the symmetry of reflection through a site (RLM type model).

From a technical point of view, we note that it would be most interesting to come up with a derivation of closed-form analytical expressions for the functions $\delta S(s)$ and $\delta \mathcal{F}(s)$. This would presumably require an extension of the derivation [11] of $c_{\text{eff}}$ in the presence of a boundary.

Finally, we note that the case of a conformal defect with a free parameter is very special to the non-interacting case. A modified link in an XXZ chain, for instance, would lead to a quite different behavior, with, depending on the sign of the anisotropic interaction, either an RG flow towards a uniform chain, or a flow to a chain split into two separate parts. In both cases, an analysis similar to the one in section 5 of this paper could be carried out, using e.g. density matrix renormalization group [34–36] techniques.

In conclusion, let us mention that, although we have focused solely on subsystems near a boundary in this paper, entanglement and fluctuation differences in a periodic system with

two impurities at either border of the subsystem also exhibit finite terms of $\mathcal{O}(1)$[24]. We hope to discuss this elsewhere.

**Acknowledgments:** We thank Pasquale Calabrese, Paola Ruggiero, Felix Palm, Ananda Roy and Fabian Grusdt for valuable discussions. H. Saleur was supported by the European Research Council (ERC) Advanced Grant NuQFT. H. Schlömer acknowledges funding by the Deutsche Forschungsgemeinschaft (DFG, German Research Foundation) under Germany's Excellence Strategy—EXC-2111—390814868 and by the ERC under the European Union's Horizon 2020 research and innovation programme (grant agreement number 948141). H.S. and C.T. contributed equally to this work.

# Appendix

## Perturbation theory for $\lambda \approx 1$

To further study the behavior of the slope of $\delta\mathcal{F}$ close to $\lambda = 1$ presented in Sec 3.2, we can perturb the exact plane wave solution of the homogeneous chain with open boundaries, given by

$$|k^{(0)}\rangle = \mathcal{C} \sum_j \sin(kj)|j\rangle \,, \tag{97}$$

with $|j\rangle$ the tight-binding basis and $\mathcal{C} = \sqrt{2/(L+1)}$. The corresponding energies read

$$E_k^{(0)} = -2t\cos(k), \quad k = \frac{\pi l}{L+1} \quad \text{with} \quad l = 1,\dots,L \,, \tag{98}$$

where we see that – in contrast to periodic boundary conditions – the states are non-degenerate. Hence, when introducing a perturbing potential $\tilde{V}(\lambda)$,

$$\tilde{V}(\lambda) = (1-\lambda)\,\underbrace{t\big\{|\ell+1\rangle\langle\ell| + \text{H.c.}\big\}}_{=V}\,, \tag{99}$$

we can perform the standard first order perturbation theory for systems with non-degenerate spectra. In first order, the corrected energies read

$$\begin{aligned}
E_k &= E_k^{(0)} + (1-\lambda)E_k^{(1)} = -2t\cos(k) + (1-\lambda)\langle k^{(0)}|V|k^{(0)}\rangle \\
&= -2t\cos(k) + (1-\lambda)\sin[k(\ell+1)]\sin[k\ell] \,.
\end{aligned} \tag{100}$$

Perturbation of the standing waves leads to

$$|k\rangle = |k^{(0)}\rangle + (1-\lambda)|k^{(1)}\rangle = |k^{(0)}\rangle + (1-\lambda)\sum_{k_1\neq k}\frac{\langle k_1^{(0)}|V|k^{(0)}\rangle}{E_k^0 - E_{k_1}^{(0)}}|k_1^{(0)}\rangle \,. \tag{101}$$

For the correlator $G_{ij} = \langle c_i^\dagger c_j\rangle = \sum_{k<k_F}\langle i|k\rangle\langle k|j\rangle$ then follows in linear order $(1-\lambda)$,

$$G_{ij} = \underbrace{\sum_{k<k_F}\langle i|k^{(0)}\rangle\langle k^{(0)}|j\rangle}_{=G_{ij}^{(0)}} + (1-\lambda)\underbrace{\sum_{k<k_F}\langle i|k^{(1)}\rangle\langle k^{(0)}|j\rangle + \langle i|k^{(0)}\rangle\langle k^{(1)}|j\rangle}_{G_{ij}^{(1)}} \,. \tag{102}$$

---

[24]This holds also when the two impurities are separated far away from each other and are hence "non-interacting".

Here,

$$\langle i|k^{(0)}\rangle = \mathcal{C}\sin(ki),$$

$$\langle i|k^{(1)}\rangle = \mathcal{C}^3 \sum_{k_1 \neq k} \frac{\sin[k_1(\ell+1)]\sin[k\ell] + \sin[k(N_{\text{imp}}+1)]\sin[k_1\ell]}{2[\cos(k_1)-\cos(k)]}\sin(k_1 i). \tag{103}$$

For computing the difference of the fluctuations between even/odd scenarios, we will consider the two perturbing potentials $V^e, V^o$, resulting in different perturbed states and correlation functions $G_{ij}^{e/o}$. The fluctuations at half filling are given by

$$\mathcal{F}(\ell) = \frac{\ell}{2} - \sum_{\substack{i,j=1 \\ i \neq j}}^{\ell} G_{ij}^2. \tag{104}$$

The first order correction in $(1-\lambda)$ to $\Delta\mathcal{F} = \delta\mathcal{F}^{(0)} + (1-\lambda)\delta\mathcal{F}^{(1)}$ hence reads

$$\delta\mathcal{F}^{(1)} = 2\left\{ \sum_{\substack{i,j=1 \\ i \neq j}}^{\ell^o} G_{ij}^{(0)}G_{ij,o}^{(1)} - \sum_{\substack{i,j=1 \\ i \neq j}}^{\ell^e} G_{ij}^{(0)}G_{ij,e}^{(1)} \right\}. \tag{105}$$

Using Eqs. (102) & (103), Eq. (105) can be evaluated numerically. To compare with the results presented above, we compute the slopes $\Delta\mathcal{F}^{(1)}$ for different chain lengths and different impurity bond positions (i.e. at 1/2 and 1/3 of the chain), and extrapolate the values for $L \to \infty$. We find that

$$\begin{aligned} \text{Impurity at } L/2: &\quad 0.27138, \\ \text{Impurity at } L/3: &\quad 0.27315, \end{aligned} \tag{106}$$

which is in correspondence with the predictions presented in the main text, Sec. 3.2.

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
