# Peer review of "Parity effects and universal terms of O(1) in the entanglement near a boundary"

_SciPost Physics, doi:SciPost Phys. 13, 110 (2022)_

## Round 1 · Referee Report · Anonymous (Referee 1) · 2022-7-28

Strengths

  1. Originality
  2. Careful and deep analysis of the physical mechanism.

Weaknesses

  1. The analytical section is probably too concise and it may look obscure for a nonexpert on the subject.

Report

This work considers the universal parity effects of the entanglement of critical chains in the presence of both boundaries and impurities.
It is found that the entanglement entropy of an interval starting at a boundary point and terminating at a defect displays oscillations depending
on the parity of the length of the subsystem. This feature, which is the main result of the work, is understood both analytically and numerically,
and it is pointed out that its universal origin relies on a scattering phase at the Fermi sea.
I consider the work valuable, deserving possibly further investigation, and I recommend it for SciPost.

Requested changes

Some minor remarks: 1. It is not completely clear how the authors derived the result Eq. 42. It seems that only the most relevant operator, different from the identity, has been kept in the expansion (which is a scalar vertex operator). Probably a small comment regarding the approximation involved and the motivation behind is needed, being Eq. 42 the core of the analytical analysis. 2. Before Eq. 44, it would be better to emphasize the fact that the entropy of the interval (0,\ell) is computed. Also, in Eq. 46 it would be nice to write down explicitly the transformation law of the vertex operator which relates the two geometries. 3. After Eq. 39. The "cocycles" \eta_{R,L} are usually called "Klein factors". A brief comment about it, just to improve readability, would be appreciated.

Minor typos: 4. Eq. 1: there is a minus sign missing 5. Page 13. Before Eq.44. It is written "This involves a p-sheeted... ... coordinates (x=L,\tau =0)". The replacement L-> \ell is needed. 6. Page 14. After Eq. 52 replacement L-> \ell. Eq. 32 "cos" -> "\cos" 7. Page 15. Eq. 58 a "dx" is missing. 8. Page 19. Eq. 83. There is an additional ")" before \ket{j}

---

## Round 1 · Referee Report · Anonymous (Referee 2) · 2022-8-15

Strengths

1- Convincing numerical and analytical results. 2- Surprising presence of a 'universal' term.

Weaknesses

1-The presentation is in a few places hard to follow.

Report

The authors provide convincing evidence, both numerically as well as analytically, for a novel universal term of O(1) in the bipartite entanglement of XX chains appearing in the presence of an impurity at the end of the subsystem.

I consider the research to be of high quality and the results very interesting. Several avenues for further exploration are clearly present. I only have relatively minor comments related to the presentation of an otherwise well written manuscript. I recommend publication after considering these minor revisions.

Requested changes

1- The manuscript could benefit from clarifying if the authors 'conjecture' the universal term or if they 'prove' that the term is universal. Given the perturbative nature of the result, Eq. (57), I believe it is the former. However, the text is in some cases confusing. For instance, the abstract mentions 'exhibit universal features'. In the box below Eq (7), the authors 'claim' the term is universal. Why not in both cases point out it is a conjecture (albeit with overwhelming evidence). 2- It is not clear to me what the authors mean by "topological phase precursor" on page 23. Is it possible to elaborate ? 3- The impurity strength is discussed in terms of $J'$,$\lambda$,$s$ and $\mu$. For the casual reader it would have been helpful to just have used one measure which probably should have been $\lambda-1$. For instance, Eq (19) uses, $\lambda -1$, where as the essentially identical Eq (41) is written in terms of $(J'-J)$. 4. Just above Eq. (51) I believe the is should be $R_p(\mu)/R_p(\mu=0)$ instead of $R_p(\lambda)/R_p(\lambda=0)$

---

## Round 1 · Referee Report · Anonymous (Referee 3) · 2022-9-6

Strengths

nondecaying parity effects is a novel feature

Weaknesses

some interpretations may be oversimplified

Report

The manuscript discusses parity effects in free-fermion chains with various types of defects. It is shown that the entropy difference between even/odd segments neighboring the defect acquires a finite value that does not decay with the length of the segment. Similar behaviour is found for the particle number fluctuations. For a single defect, the functional dependence on the impurity strength is analyzed both numerically, and a perturbative treatment using bosonization techniques is also provided for weak impurities. The setup is also generalized to extended defects, with the modified links arranged in an alternating structure. It is shown that the parity effects depend only on the phase shift at the Fermi level. Finally, an impurity of quantum dot type is also discussed, which exhibits the physics of healing, with the parity terms showing a nontrivial RG flow towards a pure system.

The results presented are novel and interesting, and I would recommend the publication of the manuscript with the following comments for the authors to address.

Requested changes

1) I'm not sure about the correctness of the interpretation presented in Table 1. The figure seems to suggest that the entanglement for weak/strong defects results from a valence bond contribution. However, considering the odd case at $\lambda=0$, one has an exact zero mode for both left/right parts of the odd chains around the cut. Introducing a weak defect should somehow couple these zero modes (which are extended states in both the left/right hand side chains) yielding a ln(2) contribution in the entropy. This is, however, a different type of contribution as that for $\lambda \gg 1$, where a valence bond is indeed formed around the defect. I have thus the feeling that the picture is oversimplified as it suggests that all the entanglement structure in the ground state is encoded by simple valence bonds along the chain, which is certainly not true. In fact, the cartoon is rather reminiscent of a ground state in a random singlet phase.

2) Below Fig. 6 the authors address the question, to what extent the pure SSH model differs from the one where the impurity is attached to a metallic piece. This is exactly the question that was studied in arXiv:1503.09116. One could also mention the related study arXiv:1406.7832 about the entanglement properties of SSH chains.

Typos:

  • In the inset of Fig. 3b: $C$ should read $C_{eff}$
  • Sentence before Eq. (51): the argument of $R_p$ should be $\mu$

---

## Round 2 · Author Response

We would like to thank all three referees for their careful reading and helpful suggestions for improving the manuscript. Below is a collection of responses addressing the raised remarks by the referees.

Referee 1:
We have added additional comments to the analytical discussion, which hopefully clarify some subtleties and improve the overall readability of the section.

Referee 2:
We slightly reformulated our conjecture, hopefully clarifying the discussion of the universality of terms of O(1).

In Chapter 4, we have pointed out that entanglement entropy differences in the full SSH model share similarities with the case of an extended alternating impurity as well as a single modified bond. Though the underlying models are fundamentally different (i.e. while the SSH model is gapped and features a topologically non-trivial phase, the impurity system is gapless), we consider the resemblance of the terms of O(1) in topological and closely related non-topological models to be an intriguing open possibility for further investigations.

Referee 3:
We agree with the referee that the valence bond picture does not capture the exact nature of the ground state. Nevertheless, it yields an intuitive platform to understand the entanglement entropy differences in the limit of weak and strong impurities, which we have clarified in the revised manuscript.

Lastly, we thank all referees for pointing out various typos, notation inconsistencies and literature suggestions, which we have addressed in the revised version.

---

## Editorial Decision

published